# ZBTB18 regulates cytokine expression and affects microglia/macrophage recruitment and commitment in glioblastoma

Roberto Ferrarese[1,2,10], Kevin Joseph [1,10], Geoffroy Andrieux [3,10], Ira Verena Haase[1], Francesca Zanon[1], Eva Kling[1], Annalisa Izzo [1], Eyleen Corrales [3,4], Marius Schwabenland [5,6,7], Marco Prinz [5,6,7], Vidhya Madapusi Ravi [1], Melanie Boerries [3,4,8], Dieter Henrik Heiland [1,8] & Maria Stella Carro [1,9] ✉

Glioma associated macrophages/microglia (GAMs) play an important role in glioblastoma (GBM) progression, due to their massive recruitment to the tumor site and polarization to a tumor promoting phenotype. GAMs secrete a variety of cytokines, which facilitate tumor cell growth and invasion, and prevent other immune cells from mounting an immune response against the tumor. Here, we demonstrate that zinc finger and BTB containing domain 18 (ZBTB18), a transcriptional repressor with tumor suppressive function in glioblastoma, impairs the production of key cytokines, which function as chemoattractant for GAMs. Consistently, we observe a reduced migration of GAMs when ZBTB18 is expressed by glioblastoma cells, both in cell culture and in vivo experiments. Moreover, RNA sequencing analysis shows that the presence of ZBTB18 in glioblastoma cells alters the commitment of conditioned microglia, suggesting the loss of the immune-suppressive phenotype and the acquisition of pro-inflammatory features. Thus, therapeutic approaches to increase ZBTB18 expression in GBM cells could represent an effective adjuvant to immune therapy in GBM.

Glioblastoma (GBM) is the most common and aggressive form of brain tumor in adults, characterized by poor prognosis, and resistant to aggressive therapies[1]. Over the past decades, gene expression profiling of GBM bulk tumors and, more recently, single cell analysis, have contributed to the identification of multiple GBM subclasses and GBM cell states with distinct molecular characteristics[2–6]. It is now evident that GBM are very heterogeneous, and that more than one cell state can co-exist in the tumor; moreover, the tumor microenvironment can affect the transitioning through different states[6]. Of note, GBM re-occurrence is accompanied by a shift to a mesenchymal state[7], which is associated with radiation resistance and reduced survival[4,8].

Glioma-associated macrophages (GAMs) constitute more than 30% of infiltrating cells in GBM and have been shown to be able to control cancer initiation, growth, and response to therapy[9]. GAM populations include primary immune cells of the central nervous system (microglia), which originate from erythro-myeloid progenitors in the yolk sac, and bone marrow-derived circulating macrophages (here referred to as macrophages), which are recruited to the tumor site[10,11]. Microglia are highly dynamic cells, which shift through different states in response to external stimuli, including the microenvironment, both in physiological and pathological conditions, ultimately affecting their function[12,13]. One key mechanism by which GAMs contribute to cancer progression is by

[1]Department of Neurosurgery, Medical Center-University of Freiburg, Freiburg, Germany. [2]Laboratory of General Pathology and Immunology, University of Insubria, Varese, Italy. [3]Institute of Medical Bioinformatics and Systems Medicine, Medical Center-University of Freiburg, Faculty of Medicine, University of Freiburg, Freiburg, Germany. [4]Institute of Molecular Medicine and Cell Research, University of Freiburg, Freiburg, Germany. [5]Institute of Neuropathology, Medical Center-University of Freiburg, Faculty of Medicine, University of Freiburg, Freiburg, Germany. [6]Signaling Research Centres BIOSS and CIBSS, University of Freiburg, Freiburg, Germany. [7]Center for NeuroModulation (NeuroModul), University of Freiburg, Freiburg, Germany. [8]German Cancer Consortium (DKTK), Partner site Freiburg, a partnership between DKFZ and Medical Center, University of Freiburg, Freiburg, Germany. [9]Present address: Laboratory of General Pathology and Immunology, University of Insubria, Varese, Italy. [10]These authors contributed equally: Roberto Ferrarese, Kevin Joseph, Geoffroy Andrieux. ✉e-mail: mariastella.carro@uninsubria.it

suppressing the tumor-specific immunity, which in turn favors tumor expansion[9]. Interestingly, mesenchymal transition in GBM strongly correlates with GAM infiltration[4,14], mainly due to GAM's ability to release a family of cytokines, which mediate mesenchymal differentiation in an NF-κB-dependent manner[2,3,15]. This suggests that the tumor microenvironment plays a central role in the modulation of the immune response against tumors. However, the mechanisms by which the tumor microenvironment affect GAMs and tumor-specific immunity are not completely understood. In consideration of the rising importance of immunotherapy in cancer treatment, defining the mechanisms through which tumors control GAMs has very important basic and clinical implications.

Others and our previous studies have described a role for the transcriptional repressor ZBTB18 as a negative regulator of the mesenchymal signature, which is associated with poor survival in GBM[16,17]. ZBTB18 is a member of the Broad complex or poxvirus and zinc finger (BTB/POZ-ZF) protein family, which contains several proteins involved in development and/or cancer formation. ZBTB18 is highly expressed in post mitotic neurons but lost or weakly expressed in GBM cells; its re-expression in these cells results in compromised cell proliferation and tumor growth in vivo[16,18]. Recently, we have identified a short ZBTB18 protein product (ZBTB18 Nte-SF, around 30 kDa) that is specifically generated by proteolytic cleavage of the full-length protein in GBM cell lines, but not in normal cortical tissues[16,19]. We have also provided evidence that ZBTB18 Nte-SF, by positively affecting HIF1A target gene expression, acquires tumor promoting capabilities, which are opposite to the tumor suppressor role of full-length ZBTB18[19].

Here, we show that ZBTB18 expression in GBM cells halts the production of key cytokines, required to recruit GAMs to the tumor site. In addition, microglia conditioned by ZBTB18 expressing GBM cells show commitment towards a pro-inflammatory, rather than a pro-tumorigenic phenotype, as it happens when normal GBM cells are used for the conditioning. Finally, the effects of ZBTB18 on the regulation of the immune response seem to be specifically associated with the full-length protein and not with the N-terminal short variant.

## Results

### Full-length ZBTB18 represses cytokine expression in GBM cells
In order to investigate whether re-expression of ZBTB18 in GBM cells is able to affect their ability to communicate with the tumor microenvironment, we examined previously (BTSC233, JX6, and SNB19)[16,20] and newly (BTSC268 and BTSC475) generated gene expression data obtained upon ZBTB18 expression in GBM cells (Supplementary Fig. 1a, b). The data revealed that ZBTB18 is a putative transcriptional repressor of several secreted factor and cytokine genes, namely CX3CL1, CCL2, CXCL14, CMTM7, S100A6, CXXC5, PDGFRA and GDF15, which appeared to be downregulated in at least four out of five experiments (Fig. 1a, Supplementary Fig. 1c). We decided to focus on CX3CL1, CCL2 and GDF15, which are known to affect GAM recruitment and/or commitment, and T cell functions[21,22]. Real-time qPCR analyses confirmed that the ectopic expression of ZBTB18 in a GBM cell line (SNB19) and in the primary GBM stem cell line BTSC233 downregulated CCL2, GDF15, and CX3CL1 genes (Fig. 1b). Next, we assessed whether ZBTB18 also affected cytokine availability by measuring the level of cytokines secreted into the culture medium by the GBM cells. Thus, we tested by cytokine antibody arrays the medium conditioned by BTSC233, transduced with the empty vector (EV) or ectopically expressing ZBTB18. Overall MCP1/CCL2 stood out as specifically and strongly reduced by the presence of ZBTB18 compared to the EV control; conversely, CX3CL1, which had also shown downregulation at the gene expression level, did not show any change across the different samples (Fig. 1c, d). The results were further confirmed in SNB19 cells (Supplementary Fig. 1d, f). Of note, only ZBTB18 full length (FL) and not the previously described pro-tumorigenic N-terminal short variant (ZBTB18 Nte-SF)[19] led to a reduced CCL2 secretion when the two ZBTB18 forms were expressed in BTSC233 (Supplementary Fig. 1e, f). Conversely, two cytokines, IL8 and angiogenin were consistently upregulated (Fig. 1c, d), although no change in the corresponding transcripts had been detected in

our analysis (Supplementary Fig. 1c). GDF-15, which our gene expression data suggested to be also regulated by ZBTB18, was not included in the cytokine array; thus, we performed an ELISA to assess its concentration in the medium conditioned by SNB19 or BTSC233 cells. Consistently with the previous data, GDF-15 concentration was reduced upon ZBTB18 FL ectopic expression in both cell lines (Fig. 1e).

In silico analyses using GlioVis (http://gliovis.bioinfo.cnio.es/)[23] evidenced a positive correlation between the expression of CCL2, GDF15 and ANG (angiogenin) and the CNS WHO grade of the brain tumors (Supplementary Fig. 2a); conversely, a negative correlation was found between ZBTB18 and CCL2 (r = −0.50, p = 0.00) and GDF15 (r = −0.62, p = 0.00) (Supplementary Fig. 3a, c). A negative correlation was also observed between ZBTB18 and ANG (r = −0.61, p = 0.00), which seems to contradict the outcome of the cytokine array (Supplementary Fig. 3a, c). CXCL8 (IL8) gene was not present in the database and could not be confirmed. Moreover, GBMs with high CCL2 and ANG, but not with GDF15 expression were associated with a worse patients' prognosis (Supplementary Fig. 2b). The negative correlation between ZBTB18 and these cytokines was detected even within a dataset including GBM only; however, since ZBTB18 is almost uniformly absent from GBM, the statistical power of these analyses is consequently weaker. Within this dataset, we observed a uniformly low expression of ZBTB18, compared to different levels of cytokine expression (Supplementary Fig. 3b, c). A similar negative correlation between ZBTB18 and CCL2, GDF15, and ANG was observed also using the cBioPortal Platform[24] (Supplementary Fig. 4a). Expanding the cBioPortal analysis, ZBTB18 showed the tendency to be mutually exclusive with CCL2, GDF15, CXCL8, and ANG (Supplementary Fig. 4b, c).

The previously described role of ZBTB18 as a transcriptional repressor suggests that ZBTB18 might be directly involved in CCL2 and GDF15 repression. Therefore, we searched for ZBTB18 (ZNF238) motifs in motifDB[25] and found a few quite consistent motifs in several databases (Supplementary Fig. 5a). Counting for the occurrence of each motive with at least 90% homology in a region ±500 bp around TSS of selected genes led to the identification of the hPDI and HOCOMOCO motifs in almost all transcripts (Supplementary Fig. 5a). Previous studies from others and us indicate that ZBTB18 binds in proximity to the TSS of its target genes[20,26]. Therefore, we focused on the predicted sites, which are located close to the transcription start (105 bp upstream the TSS (GDF15) and 365 bp downstream (CCL2)) (Supplementary Fig. 5b). Chromatin immunoprecipitation (ChIP) in BTSC233 cells confirmed the direct binding of ZBTB18 at the CCL2 and GDF15 promoter close to the TSS (Fig. 1f).

### ZBTB18 and LSD1 depletion differently impact CCL2 and GDF15 expression and secretion
Next, we sought to validate the role of ZBTB18 as regulator of cytokine expression and secretion by CRISPR/Cas9 knockdown (Supplementary Fig. 6a–c). Cytokine array in BTSC475-sgZBTBT18 #5(4) cells further proved the increased production of CCL2, while IL8 and angiogenin appeared downregulated, consistent with the result obtained in the ZBTB18 overexpression experiment (Fig. 2a). We then analyzed CCL2 and GDF-15 production and secretion by ELISA in two independent ZBTB18 knockdown lines established in BTSC475 cells (Supplementary Fig. 6a–c) and the previously described in BTSC475-sgZBTB18 #4(24)[20]. These data confirmed an increase, albeit modest, in CCL2 secretion (Fig. 2b, left panel). We speculated that, since the level of secreted CCL2 is very high in the control BTSC475 (as can be observed in the cytokine array in Fig. 2a), the knockdown of ZBTB18 produces only a mild effect. Conversely, GDF-15 secretion appeared reduced (Fig. 2b, right panel). However, in these conditions, a reduced occupancy by ZBTB18 at the CCL2 and GDF15 promoters was observed. (Fig. 2c).

The results collected so far suggest that ZBTB18 loss might have a different impact on CCL2 and GDF-15 expression and secretion, which required further investigation of the underlying mechanism. We have recently reported that ZBTB18 can regulate gene expression by interfering with LSD1 activity[20]. Therefore, we explored whether LSD1 silencing, alone

or in combination with ZBTB18 loss, affected *CCL2* and *GDF15* expression. Interestingly, LSD1 silencing led to *CCL2* downregulation, consistent with our previously described LSD1 activating role[20] (Fig. 2d and Supplementary Fig. 7a). Instead, *GDF15* appeared upregulated, which might suggest that the two cytokines are controlled by different LSD1-related mechanisms (Fig. 2d). ZBTB18 loss caused a slight upregulation of *CCL2*; however, the combination of ZBTB18 knockdown and LSD1 silencing did not rescue *CCL2* expression, hinting that ZBTB18 likely exerts its regulation by halting LSD1-mediated activation, as previously described for other ZBTB18 targets[20] (Fig. 2d and Supplementary Fig. 7a). Conversely, *GDF15* expression increased upon ZBTB18 or LSD1 loss and its upregulation persisted upon the concomitant knockdown of both factors, possibly showing an additive

effect (Fig. 2d and Supplementary Fig. 7a). Treatment of BTSC475 with the small LSD1 inhibitor, RN-1, also resulted in *GDF15* re-expression, further confirming LSD1 repressive role (Fig. 2e and Supplementary Fig. 7b). In order to test whether LSD1 affects the presence of histone marks, consistent with its histone demethylase activity, we performed ChIP with H3K4me2 and H3K9me2 antibodies in GBM#22 cells with LSD1 knockout, which were previously described[20,27]. The results clearly indicated an enrichment of H3K4me2 at the *GDF15* promoter only (Supplementary Fig. 7c), which is in line with the observed repressive role of LSD1 in *GDF15* regulation (Fig. 2d, e). Conversely, no H3K4me2 change was observed at the *CCL2* promoter, suggesting that *CCL2* activation does not require LSD1 histone demethylase activity, as also inferred by the lack of regulation observed upon

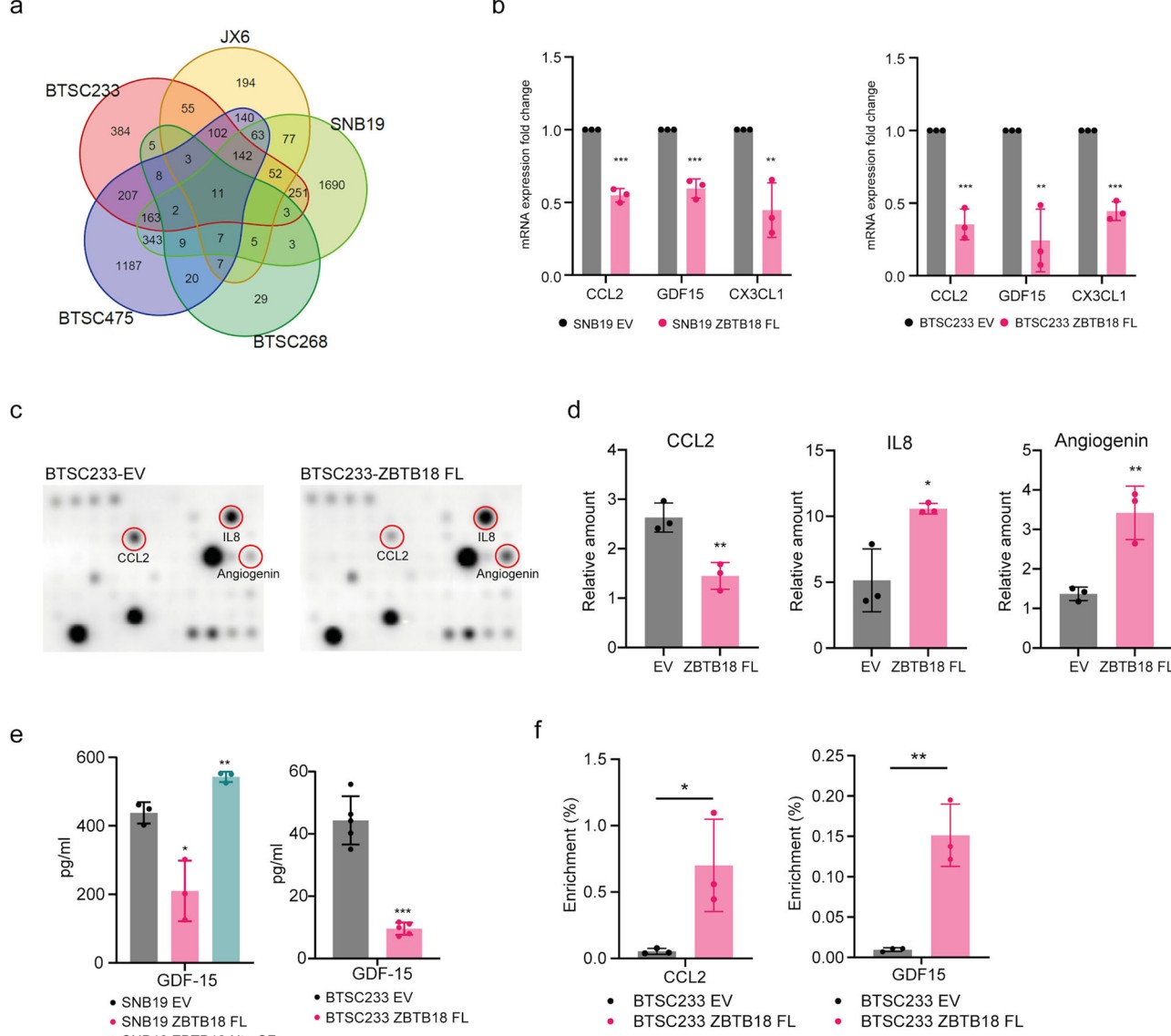

**Fig. 1 | ZBTB18 halts the expression and secretion of cytokines in GBM cells.**
**a** Venn diagram representing the intersection of three gene expression arrays performed on SNB19, BTSC233, and JX6 cells and two RNAseq analysis in BTSC268 and BTSC475, with or without ZBTB18 FL overexpression. **b** Quantitative RT-PCR of selected cytokine genes in SNB19 and BTSC233 cells with or without ZBTB18 overexpression. n = 3 biological replicates; error bars ± s.d.; *P < 0.05, **P < 0.01, and ***P < 0.001 by a t test. Gene expression was normalized to 18S RNA. **c** Representative cytokine array membranes performed on BTSC233 transduced with empty vector (EV) or ZBTB18 FL. The significantly down- or up-regulated cytokines upon ZBTB18 FL overexpression are highlighted. **d** Quantification of the

significantly regulated cytokines highlighted in (**c**). n = 3 biological replicates, error bars ± s.d.; *P < 0.05, **P < 0.01, and ***P < 0.001 by a t test. **e** GDF-15 ELISA showing the amount of GDF-15 secreted in the conditioned medium of SNB19 ectopically expressing ZBTB18 FL or ZBTB18 Nte-SF, or of BTSC233 cells upon ZBTB18 FL overexpression. n = 3 biological replicates (SNB19), n = 5 biological replicates (BTSC233); error bars ± s.d.; *P < 0.05, **P < 0.01, and ***P < 0.001 by a t test. **f** Enrichment of FLAG-ZBTB18 ChIP at the *CCL2* (left panel) and *GDF15* (right panel) promoters, in BTSC233 cells transduced with empty vector (EV) or FLAG-ZBTB18 FL. Graphs show the average q-RT PCR results of three independent ChIPs expressed in % input as indicated. Error bars ±s.d.; *P < 0.05 by a t test.

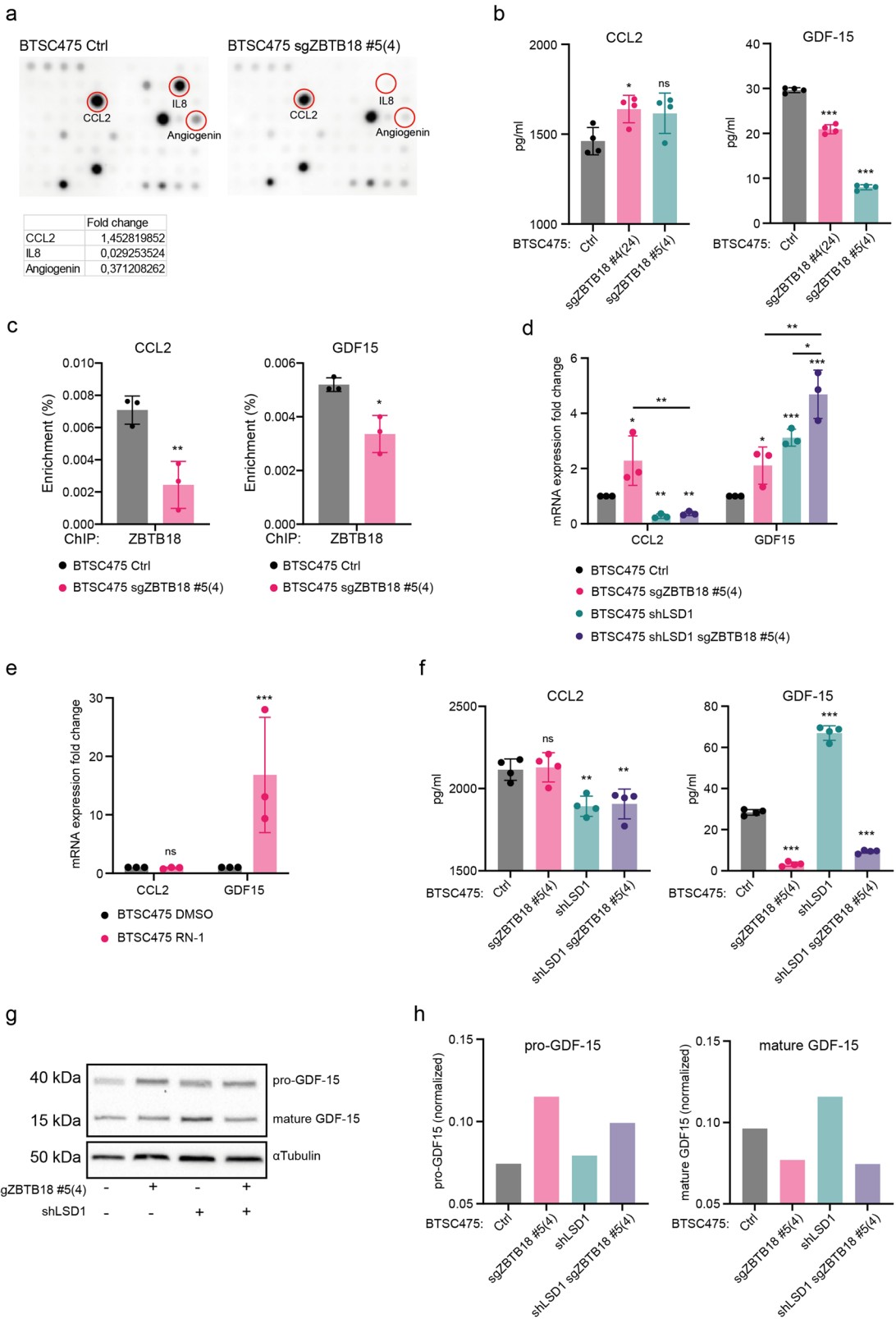

RN1 treatment (Fig. 2e and Supplementary Fig. 7b). Examination of CCL2 and GDF-15 secretion in the same experimental setting by ELISA revealed that the effect of LSD1 and ZBTB18 loss on CCL2 secretion was in line with the qPCR result (Fig. 2f, left panel), even though the concentration of CCL2 in BTSC475 Ctrl and sgZBTB18 #5(4) was very high and thus, resulting in a possible underestimation of the actual cytokine amount. Conversely, the

pattern of GDF-15 secretion did not reflect the observed gene expression changes (Fig. 2f, right panel). In fact, consistent with the previous results, less GDF-15 was released in the medium upon ZBTB18 knockdown, and the same effect lasted in presence of LSD1 silencing (Fig. 2f, right panel). The reduced secretion of GDF-15 despite its increased expression might be explained by an accumulation of the GDF-15 protein in the cell, consistent

**Fig. 2 | Removing ZBTB18 and LSD1 differentially impact CCL2 and GDF15 expression and secretion. a** Cytokine array membranes performed on BTSC475 Ctrl and BTSC475-sgZBTB18 #5(4). Relevant cytokines are highlighted. The calculated fold change based on the normalized signals is indicated. **b** CCL2 (left panel) and GDF-15 (right panel) ELISA showing the amount of secreted proteins in the conditioned medium of two independent ZBTB18 knockdown cell lines (BTSC475sgZBTB18 #4(24) and BTSC475sgZBTB18 #5(4)). n = 4 replicates; error bars ±s.d.; *P < 0.05, **P < 0.01, and ***P < 0.001 by a t test. **c** Enrichment of ZBTB18 ChIP at the *CCL2* (left panel) and *GDF15* (right panel) promoters, in BTSC475sgZBTB18 #5(4). Graphs show the average q-RT PCR results of three independent ChIPs expressed in % input as indicated. Error bars ±s.d.; *P < 0.05 by a t test. **d** q-RT PCR analysis of *CCL2* and *GDF15* expression, upon *ZBTB18* knockdown (sgZBTB18 #5(4)) and/or LSD1 silencing in BTSC475. Gene expression was

normalized to 18 s RNA. n = 3 biological replicates; error bars ±s.d.; *P < 0.05, **P < 0.01, and ***P < 0.001 by a t test. **e** q-RT PCR analysis of *CCL2* and *GDF15* expression, in BTSC475 treated with 5 μM RN-1 for 96 hours. Gene expression was normalized to 18 s RNA. n = 3 biological replicates; error bars ±s.d.; *P < 0.05, **P < 0.01, and ***P < 0.001 by a t test. **f** CCL2 (left panel) and GDF-15 (right panel) ELISA showing the amount of the secreted proteins in the conditioned medium of BTSC475 transduced with sgZBTB18 #5(4)) and/or shLSD1. n = 4 biological replicates; error bars ±s.d.; *P < 0.05, **P < 0.01, and ***P < 0.001 by a t test. **g** Western blot analysis of pro- and mature GDF-15 in BTSC475 transduced with sgZBTB18 #5(4)) and/or shLSD1. αTubulin is used as a loading control. **h** Quantification of pro- and mature GDF-15 (normalized to αTubulin) detected in (**g**).

with previous reports of intracellular GDF-15 functions[28]. Indeed, western blot analysis of cytoplasmic GDF-15 abundance showed that, while pro-GDF-15 accumulated, similarly to the qPCR result (Fig. 2g, h, left panel; Fig. 2d), the level of mature GDF-15 decreased (Fig. 2g, h, right panel). Interestingly, mature GDF-15 was more abundant upon LSD1 silencing and lost upon ZBTB18 knockdown, consistent with the ELISA data (Fig. 2g, h right panel and Fig. 2f, right panel), further suggesting that ZBTB18 might be implicated in GDF-15 processing.

Overall, our ZBTB18 knockout data confirm ZBTB18 role in CCL2 expression and consequent secretion, while GDF-15 regulation appears more complex, depending on additional factors. In particular, LSD1 seems to be implicated with different roles in the regulation of the two cytokines and it warrants further investigation.

Exploration of publicly available single cell RNA sequencing (scRNAseq, GBMap[29]) data revealed that ZBTB18 expression is localized to GBM cells that are part of the infiltrative lineage-like cells (Oligodendrocyte progenitor cell like (OPC-like) and Neuronal progenitor cell like (NPC-like cells)) (Fig. 3a–c). Further, we observed that the expression of *CCL2* and *GDF15* was mostly localized to the reactive cell types (Mesenchymal like (MES-like)), which do not express *ZBTB18*, supporting the idea that *ZBTB18* expression represses these cytokines (Fig. 3a–c). These results hold true within spatially resolved RNA sequencing data as well, where the expression of *ZBTB18* was localized to the infiltrating border of the tumor and was inversely correlated to the expression of the tested cytokines (Fig. 3d, e, Supplementary Fig. 8a). Of note, *ANG* and *CXCL8* also appeared to be mostly associated to the MES-like cell types, which do not express *ZBTB18*. (Supplementary Fig. 8b, c). A lack of association with ZBTB18 expression was also confirmed by the spatially resolved RNA sequencing data (Fig. 3d, Supplementary Fig. 8d), suggesting that the alteration of IL8 and angiogenin secretion upon ZBTB18 perturbation is not directly linked to ZBTB18 transcriptional repressive role.

Taken together, these data suggest that ZBTB18 exerts a repressive action on the expression and consequent release of cytokines that might play a relevant role in shaping the tumor microenvironment and ultimately influence the patients' clinical outcome.

## CCL2/GDF-15 repression by ZBTB18 impairs GAM recruitment to the tumor site

To assess whether the reduced expression of immunomodulatory molecules upon ZBTB18 expression affects the recruitment of immune cells in the tumor area, sections of BTSC233-derived mouse tumors generated in our former study[16] were stained for the GAM marker IBA1, which labels both microglia and macrophages. In comparison with the controls, which showed a high degree of microglia/macrophage infiltration, less IBA1-positive cells were detected in the tumors derived from BTSC233 cells expressing ZBTB18 FL (Fig. 4a, b). Our previous characterization of the examined BTSC233-EV and BTSC233-ZBTB18 FL-derived tumors indicated that ZBTB18 FL affects tumor growth and mouse survival[16]. We also confirmed ZBTB18 expression in tumor cells originated from ZBTB18 expressing cells (Supplementary Fig. 9a, b) and

since the tumor size was similar among the two groups at the time when mice were sacrificed[16], the reduction of microglia/macrophage infiltration appears to correlate with tumor progression rather than with tumor size. In order to further characterize the infiltrating immune cell population, tumor sections were co-stained for IBA1 and F4/80, a marker that labels macrophages but not microglia. Although infiltrating cells were reduced in ZBTB18 expressing tumors, the fraction of F4/80 positive cells did not change (Supplementary Fig. 10a, b). This suggests that ZBTB18 expression affected the recruitment of both microglia and macrophages. The reduced presence of IBA1-positive cells in the tumor microenvironment was also confirmed in JX6-derived mouse xenografts (Supplementary Fig. 10c). Of note, we previously reported that, similar to BTSC233, JX6 ZBTB18-derived tumors develop more slowly compared to the control, consequently affecting mouse survival[16]. We then analyzed ZBTB18 and IBA1 expression in a cohort of glioma samples belonging to different CNS WHO grades. The presence of IBA1-positive cells in the samples shows an opposite trend compared to ZBTB18 expression, even though samples where both IBA1 and ZBTB18 appear low could be detected, suggesting that other factors could be implicated in microglia/macrophages recruitment (Fig. 4c, d). Given the limited number of biopsies available for the analysis, the statistical power of the experiment was low; thus, in order to implement these results, we performed a GlioVis[23] in silico analysis to confirm the negative correlation between *ZBTB18* and *AIF1* (IBA1) (TCGA_GBM dataset: r = −0.43, p Value = 0.00; TCGA_GBM dataset: r = −0,39; p = e-9,68, Fig. 4e and Supplementary Fig. 10d). Among the cytokines regulated by ZBTB18, CCL2 is known to be released by the glioma cells and to recruit myeloid-derived suppressor cells[22]. *CCL2* also showed the strongest positive correlation to *AIF1* (r = 0.633, p < 0.001), as confirmed by GlioVis multi-correlation analysis (Supplementary Fig. 10e). On the other hand, GDF-15 is the least characterized of the ZBTB18-regulated cytokines and its role in GAM recruitment remains to be clarified. Thus, to verify whether ZBTB18-mediated *CCL2* and *GDF15* downregulation affects GAM migration, we set up an in vitro invasion assay, using as chemoattractant the conditioned media from BTSC233 expressing EV control, ZBTB18 FL, or ZBTB18 Nte-SF with or without the addition of CCL2 or GDF-15. We used SV-40 immortalized microglia cells, which have been proven to be a useful experimental model to investigate the physiopathology of microglia cells[30,31]. Microglia cells attracted by ZBTB18 FL-conditioned medium showed reduced invasive capabilities compared to those attracted by the EV control-conditioned one, confirming the role of ZBTB18 in halting GAM recruitment. However, the addition of CCL2 removed ZBTB18 FL blockade and completely restored microglia invasiveness (Fig. 4f, Supplementary Fig. 10f). In line with these data, blocking CCL2 with a specific antibody in BTSC233-conditioned medium impaired microglia invasion while neutralizing GDF-15 did not produce the same effect (Fig. 4g). The collected evidence highlights that ZBTB18 impairs the recruitment of GAMs to the tumor site by repressing the expression of cytokines with chemoattractant properties, CCL2 in particular.

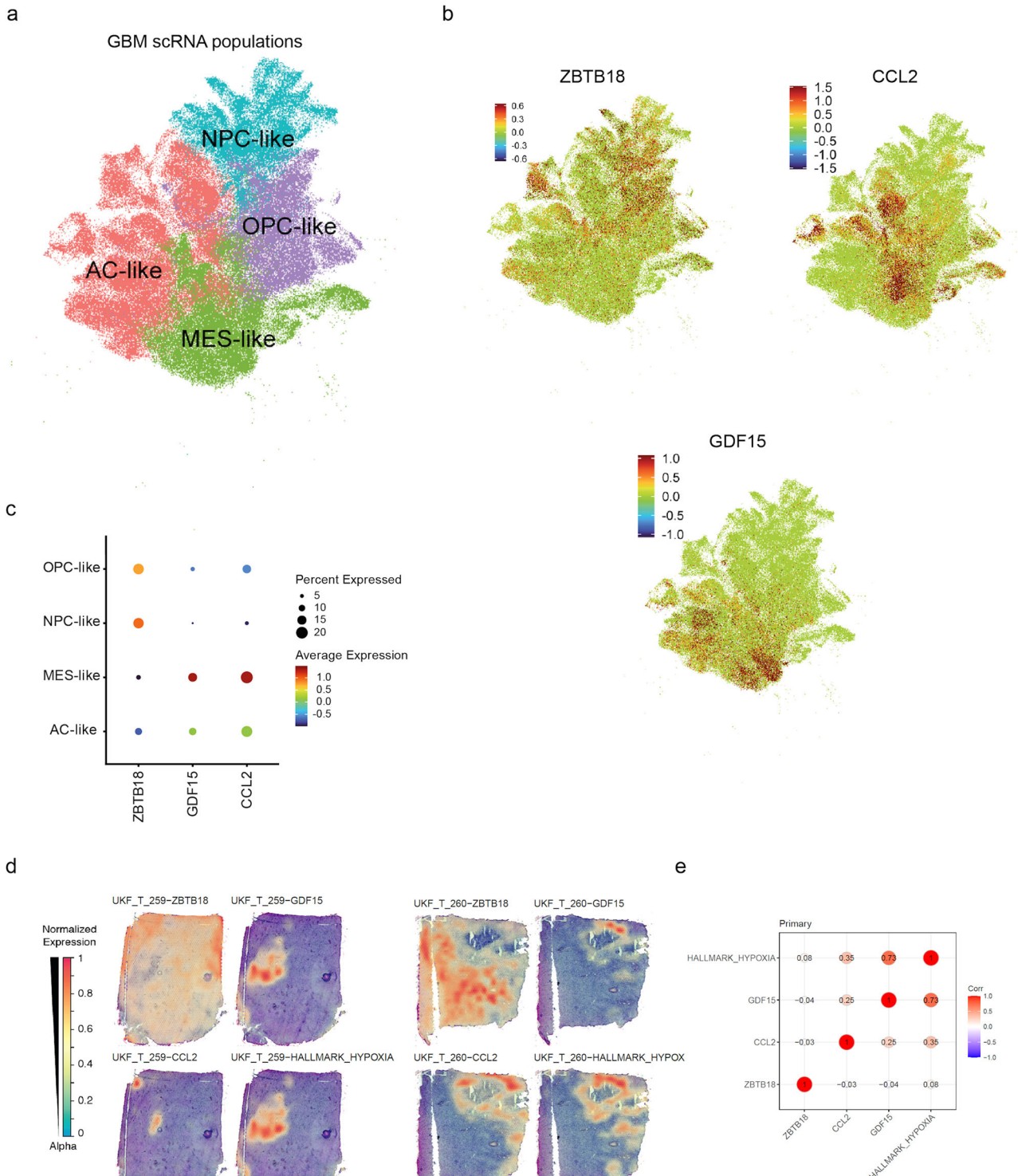

**Fig. 3 | Cytokines and ZBTB18 are complementary expressed within GBM subtypes. a** UMAP plot of the neoplastic cells within the dataset used for data analysis (GBMap). The cells are stratified based on transcriptional programs: Neuronal progenitor cell like (NPC-like), Oligodendrocyte progenitor cell like (OPC-like), Astrocytic cell like (AC-like) and Mesenchymal like (MES-like). **b** UMAP plot of *ZBTB18*, *CCL2* and *GDF15* expression within the neoplastic population. The color bar represents the expression levels represented. **c** Dotplot visualization of the enrichment of *ZBTB18*, *CCL2* and *GDF15* expression within neoplastic cell transcriptional programs. **d** stRNA visualization of the hallmark_hypoxia tumor area and the spatial localization of ZBTB18, CCL2 and GDF15 expression in two independent GBM sections. **e** Dotplot visualization of the spatial co-localization of the hallmark_hypoxia tumor area and ZBTB18, CCL2 and GDF15 expression. The correlation values are displayed.

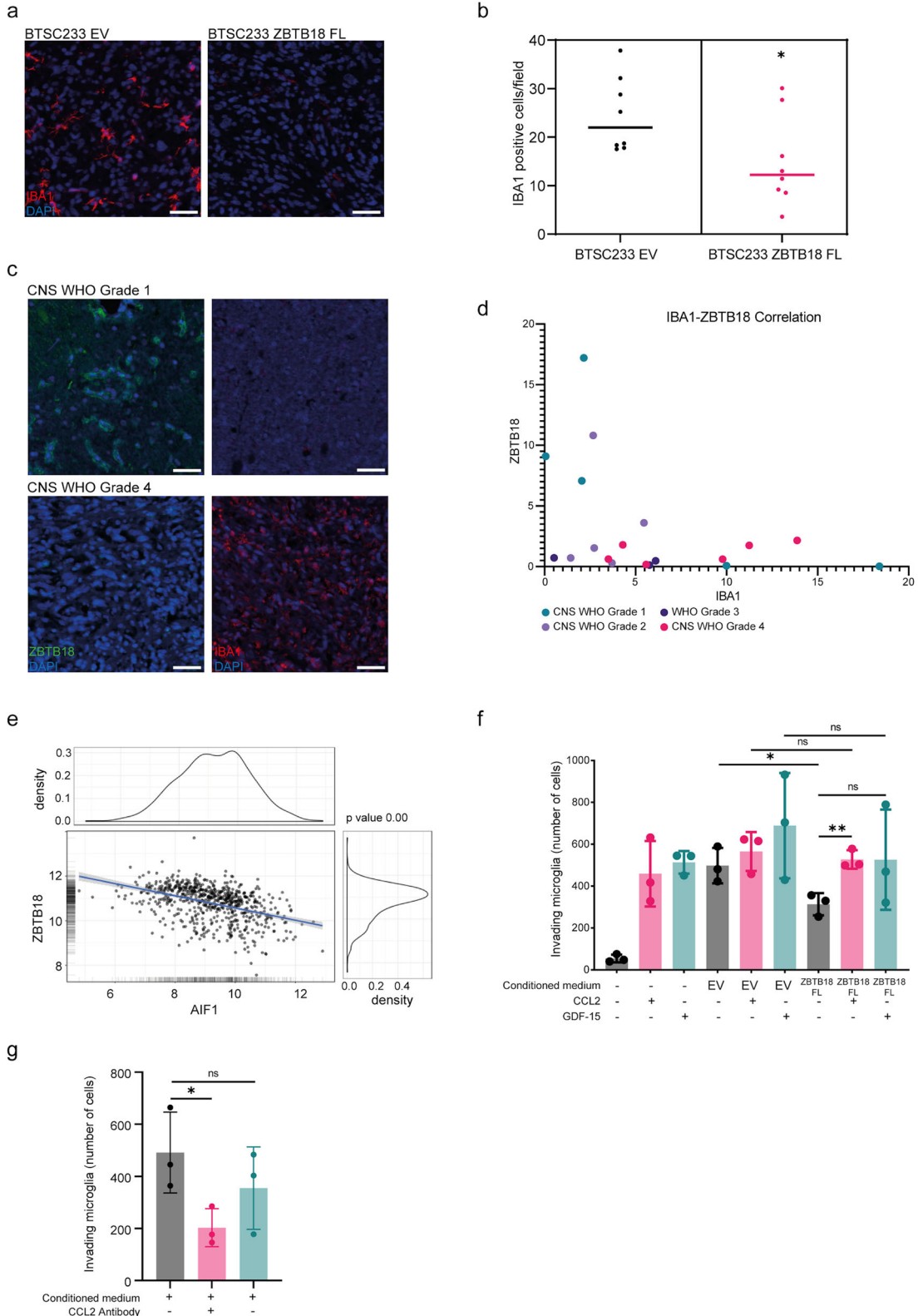

**Fig. 4 | ZBTB18 expression in GBM cells impairs the GAM recruitment.**
**a** Representative micrographs of mouse brain sections with tumors derived from BTSC233 EV or BTSC233 ZBTB18 FL xenografts and stained with IBA1 antibody; scale bar: 50 μm. **b** Quantification of the immunostaining shown in (A); n = 9 biological replicates; error bars ±s.d.; *P < 0.05 by a t test. **c** Representative micrographs of brain tumor biopsies stained with ZBTB18 (left) and IBA1 (right) antibodies; scale bar: 50 μm. **d** Correlation between IBA1 and ZBTB18 signal intensity derived from the immunostaining of brain tumor biopsies shown in (**c**); r = −0.43, p Value = 0.00.

**e** Pearson correlation analysis between ZBTB18 and AIF1 expression in the TCGA_GBMLGG dataset. **f** Invasion assay of non-treated microglia and microglia exposed to culture medium conditioned by BTSC233 EV or BTSC233 ZBTB18 FL cells. Each sample was treated with CCL2, GDF-15, or mock control. n = 3 biological replicates; error bars ±s.d.; *P < 0.05, **P < 0.01, and ***P < 0.001 by a t test.
**g** Invasion assay of microglia exposed to culture medium of BTSC233 cells treated with CCL2 or GDF-15 neutralizing antibodies. n = 3 biological replicates; error bars ±s.d.; *P < 0.05, **P < 0.01, and ***P < 0.001 by a t test.

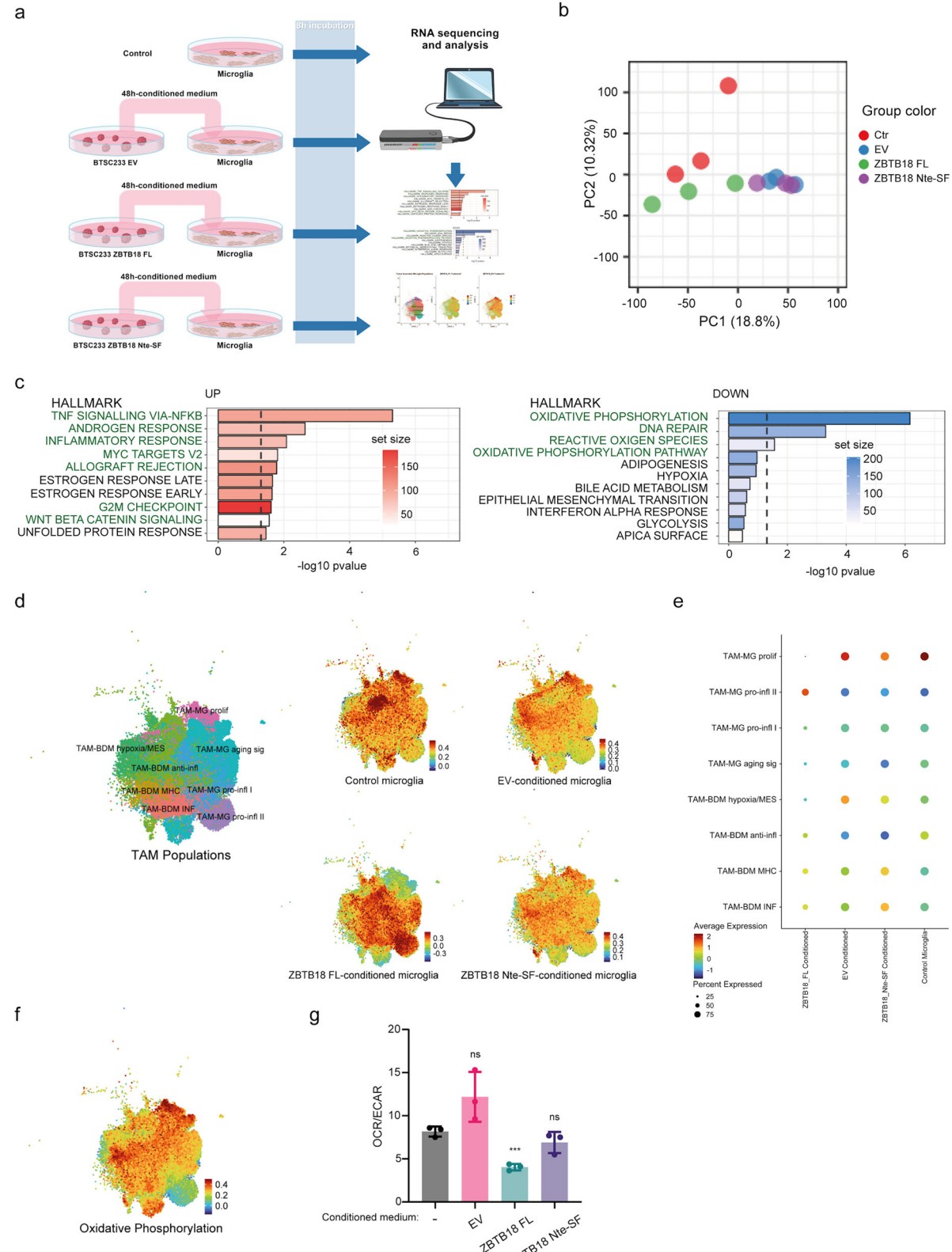

## GBM cells expressing full-length ZBTB18 induce a switch in microglia towards a more pro-inflammatory and less pro-tumorigenic phenotype

To investigate the role of ZBTB18 expressing GBM cells in driving the commitment of GAMs, we sought to characterize the transcriptome of microglia cells exposed to the medium conditioned by BTSC233 transduced with either ZBTB18 FL, ZBTB18 Nte-SF or EV. After 48 h-incubation with transfected BTSC233, the cell-free medium was added for 8 h to microglia cells, which were subsequently processed for RNA sequencing; the gene expression profiles of microglia exposed to differently conditioned media was then compared to each other and to untreated microglia as control (Ctrl) (Fig. 5a). The principal component analysis (PCA) indicated that all

**Fig. 5 | ZBTB18 expression in GBM cells affects microglia activation.**
**a** Experimental flow chart of the conditioned microglia characterization by RNAseq.
**b** Principal Component Analysis (PCA) showing the clustering of the three replicates of each experimental condition. **c** Top 10 up- or down-regulated pathways identified by Gene Set Enrichment Analysis (GSEA, gage, Hallmarks) in ZBTB18 FL-conditioned vs. EV-conditioned microglia. Processes related to pro-inflammatory or pro-tumorigenic microglia features are highlighted. **d** UMAP plot of the expression of differentially expressed gene in microglia conditioned with medium from BTSC233 expressing EV, ZBTB18 FL, or ZBTB18 Nte-SF, within the TAM population. The plots depict enrichment of cells that express upregulated genes from long read RNA sequencing mapped to the scRNA dataset. The color bar represents the expression levels of the transcriptional program represented. UMAP

plot of the TAMs within the dataset used for data analysis (GBMap) is shown as reference; the cells are stratified based on transcriptional programs. **e** Dotplot visualization of the enrichment within TAM transcriptional programs, of differentially regulated genes in microglia treated as in (**d**). **f** UMAP plot of the expression of oxidative phosphorylation genes within the TAM population. The color bar represents the expression levels of the transcriptional program represented.
**g** Oxygen consumption rate/extracellular acidification rate ratio measured by Seahorse ATP production rate assay in microglia exposed to culture medium conditioned by BTSC233 EV, BTSC233 ZBTB18 FL, or BTSC233 ZBTB18 Nte-SF cells. n = 3 biological replicates; error bars ±s.d.; *P < 0.05, **P < 0.01, and ***P < 0.001 by a t test.

conditioned microglia segregated from the control cells and that ZBTB18 FL-conditioned microglia grouped separately from the ZBTB18 Nte-SF- and EV-exposed samples, which instead clustered together (Fig. 5b). To support this finding, we made a consensus clustering partitioning using skmeans (spherical k-means clustering) clustering on top 1, 5, 10, 25, 50 and 75% genes based on ATC with 100 repetitions as described in ref. 32. With k = 2, one cluster contains samples from EV and the ZBTB18 Nte-SF and the other cluster contains samples from Ctr and ZBTB18 FL. When increasing the number of clusters, we could not identify the 4 conditions; EV and the ZBTB18 Nte-SF remained in the same cluster whereas Ctr and ZBTB18 FL were divided into sub-clusters (Supplementary Fig. 11a). In consideration of our previous data and of the differential clustering of the samples, we focused on gene changes associated with ZBTB18 FL expression compared to EV. Detailed examination of pathway and gene signatures, which in microglia associated with the conditioning by ZBTB18 FL expressing cells, pointed at the "TNFA_signaling_via_NFKB hallmark", a classical inflammatory signal, as the most enriched pathway (Fig. 5c, left panel, Supplementary Fig. 11b). This pathway also appeared highly enriched upon microglia exposure to ZBTB18 Nte-SF expressing GBM cells (Supplementary Fig. 11b). Other upregulated signatures included the "Androgen response" (Fig. 5c, left panel). Interestingly, a crosstalk between Androgen and NF-κB signaling has been described in prostate cancer, with consequent increase of the pro-inflammatory transcriptional program[33]. "Allograft rejection" is another classic inflammatory pathway, which is enriched upon ZBTB18 expression. Other upregulated pathways, such as "G2/M checkpoints" and "Myc targets" were downregulated in the recently described M4 microglia subtype, which mostly identifies myeloid-derived suppressor cells (MSCD)[13].

Notably, "Oxidative_phosphorylation hallmark" (OX-PHOS) was the top downregulated pathway in ZBTB18 FL, but not in ZBTB18 Nte-SF-exposed microglia, (Fig. 5c, right panel, Supplementary Fig. 11b). We also observed loss of expression of "DNA Repair" genes signature, which has been shown to be associated with the expression of cytokines involved in the establishment of the M2 (anti-inflammatory, pro-tumorigenic) phenotype of GBM[34]. Of note, the loss of both the OXPHOS and DNA repair signature is consistent with the increased production of free radicals during OXPHOS, with consequent DNA damage and repair[35]. When we extended our analysis to TAM scRNAseq data (GBMap), we observed that BTSC233 EV-conditioned microglia showed increased expression of genes associated to the mesenchymal-like pro-tumorigenic module TAM-BDM hypoxia/MES, while ZBTB18 FL-conditioned cells were characterized by the suppression of microglia proliferation (TAM-MG prolif) and the enrichment of pro-inflammatory genes (TAM-MG pro-infl II) (Fig. 5d, e). Conversely, we observed that OX-PHOS overlapped with proliferative microglia and that there is a loss of OX-PHOS in correspondence with the ZBTB18 FL-enriched pro-inflammatory II module (Fig. 5f). Several studies indicate that upon activation, inflammatory macrophages undergo a metabolic switch and almost exclusively rely on glycolysis for energy production[36]. We then sought to validate these findings by performing metabolic analyses on conditioned microglia. Oxidative phosphorylation and glycolysis were assessed by measuring oxygen consumption rate (OCR) and extracellular acidification rate (ECAR), respectively. Consistently with RNA sequencing

results, microglia conditioned by ZBTB18 FL expressing cells showed a lower OCR/ECAR ratio, indicating a switch towards a more glycolytic metabolism (Fig. 5g).

Examination of the most significantly altered cytokine-expressing genes in microglia cells exposed to ZBTB18 FL conditioned medium, compared to EV, revealed that several cytokine genes involved in inflammation (CXCL8, INHBA, CCL7, NTS and STC1)[37–41] were upregulated, while cytokine genes implicated in pro-tumorigenic signals, such as SEMA4C, SEMA3A and BMP6[42,43] were reduced (Fig. 6a). Among the most significantly deregulated genes, we observed downregulation of S100A4, a recently identified regulator of immunosuppressive T cells and pro-tumorigenic macrophages in GBM cells, and an important target for immunotherapy (Fig. 6a)[13]. Interestingly, OX-PHOS hallmark was shown to be enriched in immunosuppressive macrophage subtypes, in which S100A4 is also highly expressed[13]. In line with these observations, GBMap data highlight the association of S100A4 with mesenchymal GBM cells (Fig. 6b) and remarkably, with the TAM-BDM hypoxia/MES module within the TAM populations (Fig. 6c). Of note, S100A4 results to be absent from the OX-PHOS negative TAM-MG pro-infl II module, which was in turn enriched upon ZBTB18 FL-expressing cells conditioning (Figs. 5f, 6c). Finally, interrogation of pathways related to inflammation highlighted signatures of genes induced by inflammatory stimuli such as LPS, as significantly upregulated in microglia exposed to ZBTB18 FL medium (Fig. 6d, e); on the other hand, signatures related to the negative regulation of cytokines involved in inflammation were downregulated (Fig. 6d, e). Of note, there was a high degree of overlap among the co-regulated signatures, as indicated by the jaccard plots (Fig. 6e).

We then used gene set enrichment analysis (GSEA) to test whether specific myeloid signatures recently identified by scRNAseq[13] were differentially expressed in microglia cells exposed to GBM cells expressing ZBTB18. Interestingly, two pro-tumorigenic signatures, MC03 and MC04, were significantly downregulated in the ZBTB18 FL versus EV comparison, further confirming our observation (Fig. 6f). Moreover, GBMap data suggested an enrichment of the TAM-BDM MHC module in microglia conditioned with the medium of ZBTB18 FL expressing cells (Figs. 5d, 7a). This is consistent with the upregulation of HLA-DMA encoding for MHCII, a marker of pro-inflammatory microglia[44], in ZBTB18 FL expressing cells by RNA-seq (LogFC=3.47; p = 0.0012) (Fig. 6a). In order to validate these data, we observed that MHCII was more highly expressed in microglia exposed to medium conditioned by ZBTB18 FL-expressing cells, compared to EV (Fig. 7b, c).

## ZBTB18 regulates MHCII expression in a syngeneic mouse model
To test the expression of MHCII upon ZBTB18 FL expression in vivo, we first attempted to stain brain sections of the BTSC233 xenograft model (Fig. 4 and ref. 16); however, no MHCII signal was detected. We speculated that since the NOD/SCID mouse model used for the xenografts is unable to mount a proper adaptive immune response, given the absence of the T cell population, the activation of antigen presenting molecules was somehow hindered, MHCII included. Therefore, we took advantage of a recently established immunocompetent syngeneic model established in our

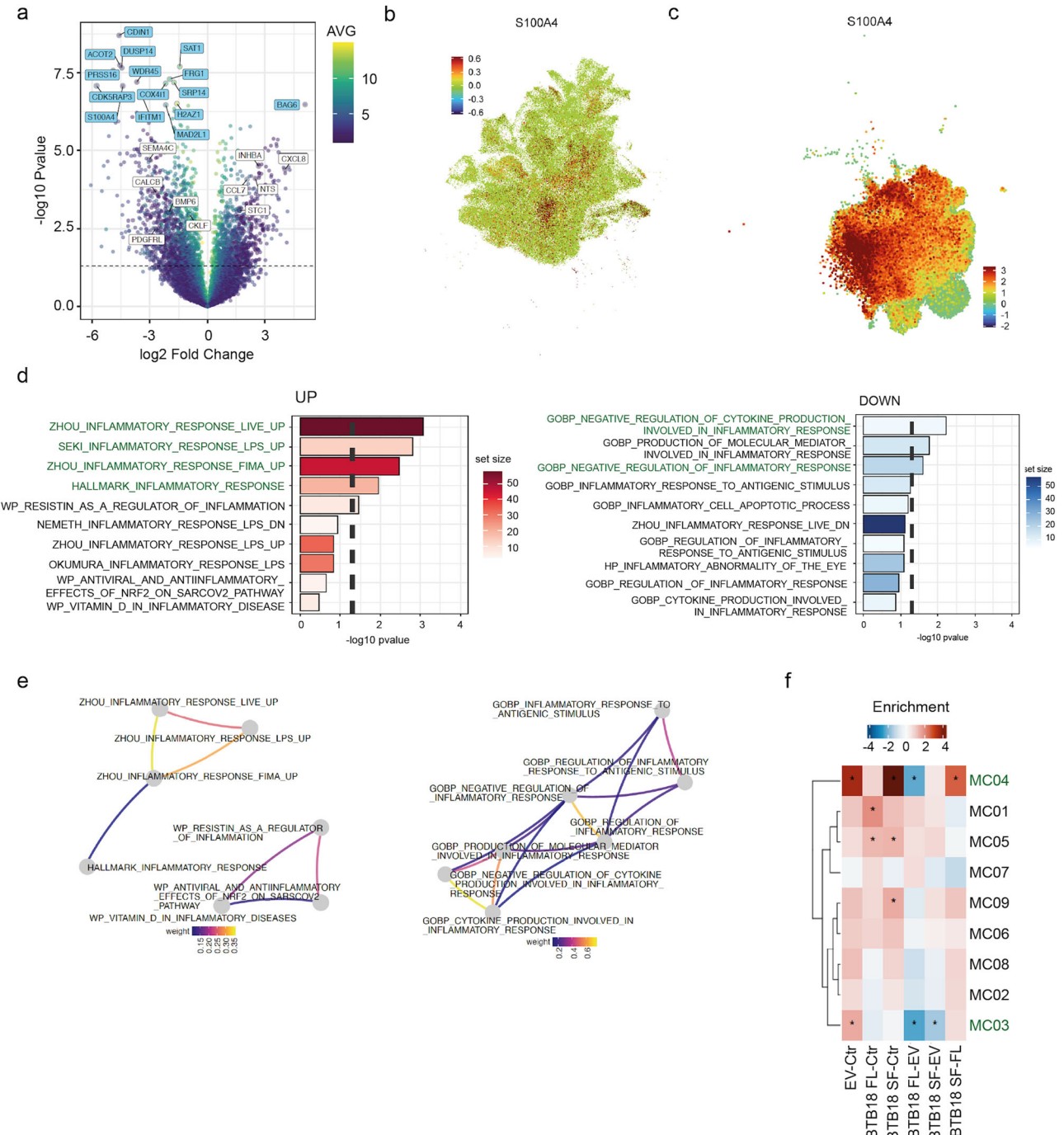

**Fig. 6 | ZBTB18 expression in GBM cells induces a shift in microglia commitment. a** Volcano plot showing the top 15 differentially regulated genes and the 10 top differentially regulated cytokines, adjusted p value (QV) in the ZBTB18 FL vs EV comparison. Selected top UP and DOWN regulated genes are labeled (light blue boxes), cytokines are marked as white boxes. Color code represents the average normalized intensity (log2). Dash line indicates the significance threshold at 0.05 adjusted p value. UMAP plot of *S100A4* expression within the neoplastic population (**b**) and within TAMs (**c**). The color bar represents the expression levels represented.

**d** Top 10 up- or down-regulated pathways identified by GSEA (gage, Keyword "Inflammation") in ZBTB18 FL-conditioned vs. EV-conditioned microglia. Processes related to pro-inflammatory or anti-inflammatory responses are highlighted. **e** Jaccard plots showing overlap between the deregulated "Inflammation" signatures. Color code represents the Jaccard index between two terms. **f** Heatmap showing the up-or downregulated myeloid signatures from Abdellfattah et al., analyzed by GSEA. Significant regulated signatures in the different group comparisons are highlighted with an asterisk (adj. p < 0.05).

laboratory. Specifically, murine GBM cells (KAB203) derived from a previously described GBM mouse model (RCAS PDGFA/shpTrp53) (Squatrito et al., personal communication[45];), were transduced with control vector or with a ZBTB18 FL expressing lentivirus and injected in C57BL/6 mice (Supplementary Fig. 12a–d). Clear solid tumors within the striatum and with GBM features were detected in 7 out of 9 mice in the EV group and

out of 9 mice in the ZBTB18 FL group, upon blind examination by a neuropathologist (Supplementary Fig. 12b). However, in some cases, tumor cells grew in the subarachnoid space outside the meninges (Supplementary Fig. 12e) causing the premature death of these animals. Thus, it was not possible to accurately determine the survival rate in the two conditions. Since one tumor in the EV group had cells growing both in the striatum and

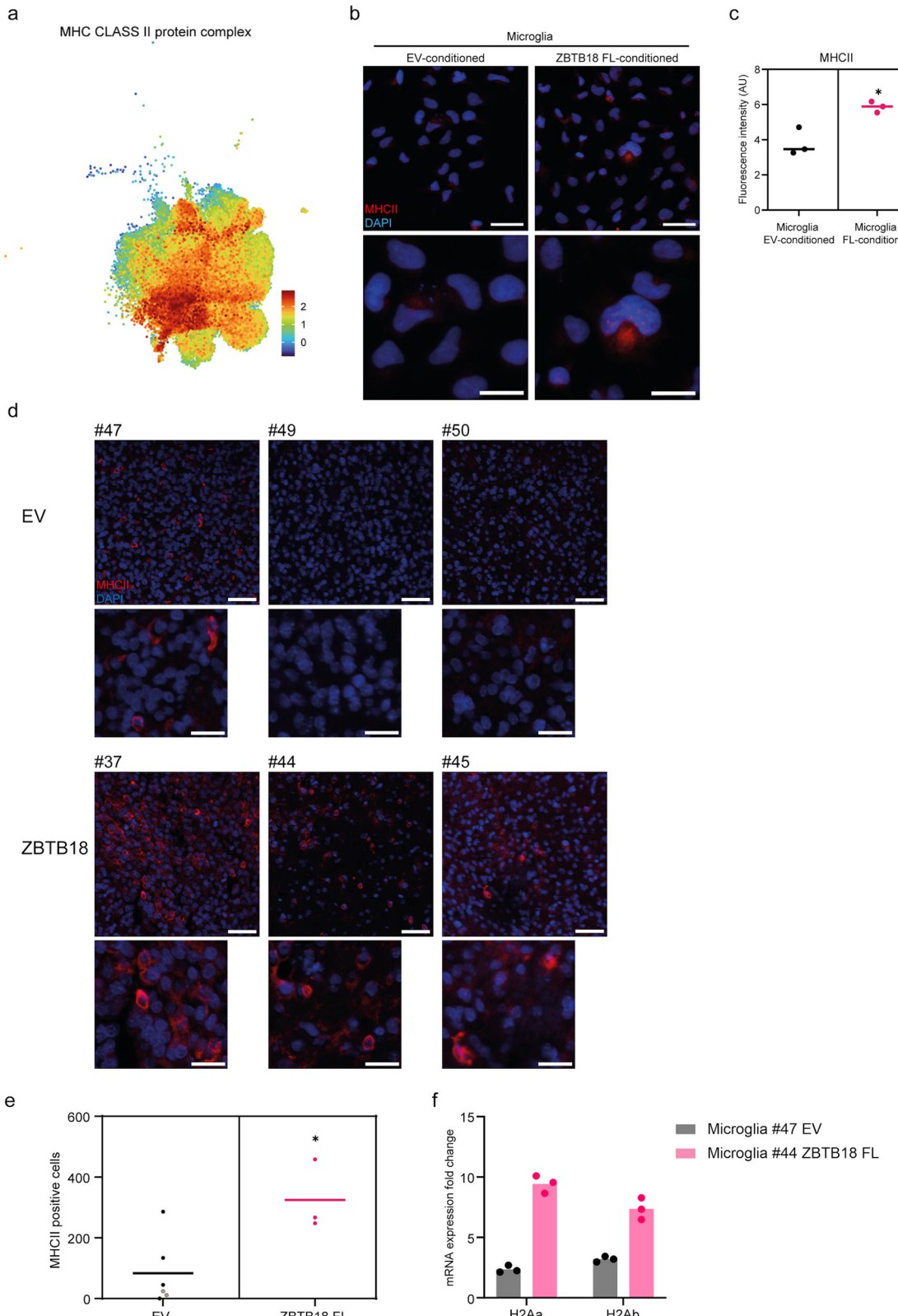

**Fig. 7 | ZBTB18 affects MHCII expression in microglia in vivo. a** UMAP plot of MHC class II protein complex expression within TAMs. The color bar represents the expression levels represented. **b** Representative micrographs of microglia conditioned with BTSC233 EV medium or BTSC233 ZBTB18 FL medium and stained with MHCII antibody; scale bar: 50 µm. **c** Quantification of the immunostaining shown in (**g**); n = 3 biological replicates; *P < 0.05 by a t test. **d** Representative micrographs of mouse brain sections with tumors derived from KAB203 EV or KAB203 ZBTB18 FL cells and stained with MHCII antibody; scale bar: 50 µm (top panels) and 25 µm (insets, bottom panels). **e** Quantification of the immunostaining shown in (**d**); n = 6 (EV), n = 3 (ZBTB18 FL) biological replicates; *P < 0.05 by a t test. Dark dots correspond to the samples shown in (**d**) and (Supplementary Fig. 12b). **f** q-RT PCR analysis of *H2Aa and H2Ab* expression, in the indicated KAB203-derived tumors. Gene expression was normalized to mouse Hprt. n = 3 technical replicates; error bars ±s.d.

outside the cortex, it was not included in the subsequent examinations. Therefore, we analyzed IBA1 and MHCII expression in 6 EV and 3 ZBTB18 FL tumors: a slight increase of IBA1 expressing cells was observed, although not statistically significant, possibly due to the limited number of samples with tumors, together with the high variability of IBA1-positive cells in ZBTB18 expressing tumors (Supplementary Fig. 12f, g). Immunostaining of MHCII indicated a higher expression in the ZBTB18 FL group (Fig. 7d, e), further supporting our previous observation. Moreover, we extracted murine microglia with the CD11b beads system (Miltenyi Biotech) followed by RNA isolation. We were able to obtain microglia RNA from two tumor samples, one EV (#47) and one expressing ZBTB18 FL (#44), which had a similar IBA1 expression pattern. Albeit only one biological replicate was available for each group, qPCR analysis showed upregulation of both HLA alleles (*H2Aa* and *H2Ab*) (Fig. 7f). This is in line with our staining and the RNAseq data obtained from conditioned microglia; however, the lack of biological replicates makes this result only preliminary and demands for additional validation.

## ZBTB18 affects pathways associated with an inflammatory switch in microglia

We then looked for transcription factors (TF) and regulatory pathways potentially involved in the control of genes differentially expressed in microglia conditioned with ZBTB18 FL medium. The Dorothea analysis identifies TCF7L2 as the most active TF based on the transcriptional regulation of its known target genes (Fig. 8a, b). TCF7L2 is a transcription factor, known to mediate the WNT pathway and associated with oligodendroglial differentiation. Interestingly, the WNT pathway has been linked to the activation of inflammatory microglia[46] and appears among the upregulated circuit unveiled by the regulatory pathway (Progeny) analysis (Fig. 8c). Here, also the MAPK, Androgen and TNF signaling pathways were enriched (Fig. 8c), in line with the GSEA results. Of note, RUNX1 and RELA, which are among the top TFs identified by Dorothea (Fig. 8a), are mediators of the MAPK and TNF pathway, respectively[47,48]. All these TOP pathways are activated in inflammatory microglia[49,50], which reinforces our idea that ZBTB18-expressing GBM cells induce an inflammatory switch. In line with this hypothesis, the major inactive TF identified by Dorothea was MEF2C, since most of its known upregulated target genes are repressed (i.e., weight > 0 and log2 foldchange < 0). This transcription factor was recently shown to restrain microglia inflammatory response[51] (Fig. 8a, b). Furthermore, Hypoxia emerged as the top downregulated regulatory pathway (Fig. 8c), consistent with recent findings indicating that pro-tumorigenic microglia is regulated by HIF1A[52].

LPS-mediated signaling in inflammatory microglia has been linked to the accumulation of lipid droplets, which are neutral lipid-rich organelles important for lipid storage and homeostasis[53,54]. Therefore, we sought to compare the lipid droplet levels in microglia conditioned with the medium collected from BTSC with ZBTB18 expression or knockdown. Microglia exposed to BTSC233 ZBTB18 FL medium had a higher lipid content compared to those conditioned with BTSC233-EV (Fig. 8d, e). Conversely, the level of lipid droplets in microglia decreased upon exposure to the medium collected from BTSC475-ZBTB18 KO (Fig. 8d, e).

Overall, our data indicate that microglia exposed to the medium of ZBTB18 expressing GBM cells lose pro-tumorigenic traits and acquire a more pro-inflammatory phenotype.

## Discussion

GBM cells have been shown to secrete several cytokines, which contribute to set a favorable microenvironment for the tumor by (1) recruiting GAMs and (2) subsequently, inducing their commitment to a pro-tumorigenic phenotype, which in turn promotes GBM growth and invasion[9]. Here, we report a new role of ZBTB18 as a repressor of the expression of cytokines implicated in the recruitment and commitment of resident microglia and circulating macrophages. We propose that the expression of ZBTB18 in GBM cells halts the production of key cytokines preventing the recruitment of GAMs to the tumor area and consequently, the establishment of the immunosuppressive microenvironment required for the tumor to expand. In particular, our data show that the transcription and secretion of MCP1/CCL2 and GDF-15 are consistently inhibited in several GBM cell lines upon ZBTB18 ectopic expression. However, the regulation of these two cytokines seems to occur through different mechanisms, which, at least in part, involve the histone demethylase LSD1. In the case of *CCL2*, ZBTB18-repressive activity likely applies through the inhibition of LSD1 activating function, similarly to what we previously described for the regulation of SREBP genes[20]. While, in the context of *GDF15* regulation, LSD1 seems to act as a repressor, independently from ZBTB18 function; in fact, *GDF15* expression was active even when the two factors were silenced, alone or in combination. Importantly, ZBTB18-loss-of function experiments revealed that, even though *GDF15* transcript increased, consistent with a reduced binding and repression by ZBTB18, the maturation of the pro-protein and its secretion were nonetheless halted. This suggests that alternative mechanisms to limit GDF-15 secretion might be activated in the tumor cell, as an adaptation to ZBTB18 reduction. The investigation of the underlying mechanism would be important, in terms of therapeutic approaches aimed at reintroducing ZBTB18 in GBM cells. In this perspective, also a deeper investigation of the complex ZBTB18 and LSD1-mediated regulatory mechanism of GDF15 and CCL2 expression, including changes of histone marks at the respective promoters, seems worth addressing in the future.

CCL2 is a well-known monocyte chemoattractant, which has been shown to recruit GAMs and contribute to GBM growth and invasion[22,55]. Numerous studies indicate that GDF-15 is a prognostic and predictive marker in various solid cancers[56]; moreover, it seems to play an important role in GBM as suggested by its high levels in the cerebrospinal fluid and association with the worst outcome of GBM patients[57]. *GDF15* silencing was shown to be associated with improved survival and increased infiltration of T-cells and macrophages in a glioblastoma syngeneic mouse model[21]. In line with previous knowledge of the identified cytokine function, our in vitro invasion experiments confirmed the role of CCL2 in microglia recruitment and positioned it downstream the regulation exerted by ZBTB18.

Nevertheless, the role of GDF-15 in immunosuppression seems less straightforward, as it does not appear as efficient as CCL2 in attracting microglia. GDF-15 might affect different immune cell types; one possibility is that ZBTB18-mediated limitation of GDF-15 secretion would impact T-cell functions. In the future, additional studies with immunocompetent models (i.e., KAB203 cells) would be required to investigate this aspect.

Cytokine arrays performed in different experimental conditions indicate that, IL8 and angiogenin, two cytokines associated with angiogenesis and poor prognosis in GBM[58,59] appeared to be more secreted upon ZBTB18 FL expression. However, none of the two genes was found to be upregulated by ZBTB18 in our gene expression analysis, nor their transcripts were found to be associated with ZBTB18 expression in Gliovis and cBioPortal databases, as well as in GBMap and spatially resolved RNA sequencing data. This propounds for an indirect ZBTB18-mediated effect or a negative feedback. Moreover, the reported role of these cytokines does not fit the previously described ZBTB18-mediated phenotype, indicating that the observed secretion might not be sufficient to induce the acquisition of more aggressive traits, which would also depend on the level of their respective receptors. In addition, it cannot be excluded that the two cytokines could be post-translationally modified and/or differentially processed upon ZBTB18 expression. Alternatively, IL8 and angiogenin might be implicated in other previously unreported functions. However, although worth further investigation, the characterization of ZBTB18-mediated regulation of IL8 and angiogenin secretion and function goes beyond the scope of this study.

Our study also indicates that ZBTB18 expression hinders the capacity of tumor cells to induce a pro-tumorigenic microglia phenotype. RNAseq analysis performed on immortalized microglia revealed that the exposure to medium conditioned by ZBTB18 FL expressing GBM cells is sufficient to induce transcriptional changes associated with loss of oxidative phosphorylation and induction of inflammatory pathways such as those mediated by LPS and TNFα/NFκB. Since our assays indicate that ZBTB18 affects the release of CCL2 and, to some extent, GDF-15, it is likely that the

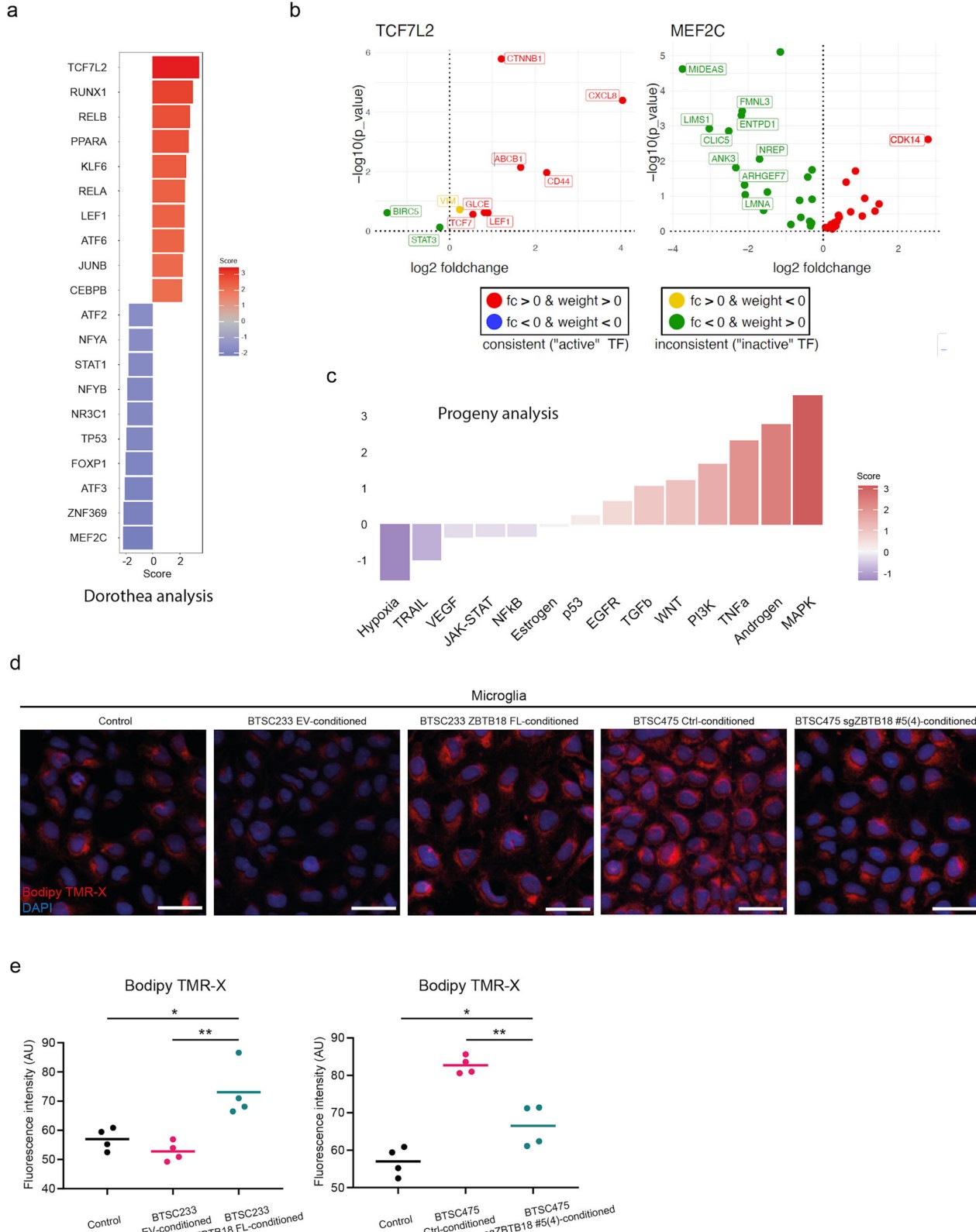

**Fig. 8 | ZBTB18 expression in GBM cells induces inflammatory-related pathways and features. a** Bar plot showing the top 10 UP and DOWN TFs associated with genes, which are deregulated in ZBTB18 FL-conditioned vs. EV-conditioned microglia, based on DOROTHEA analysis. Color code represents the score from decoupleR enrichment analysis (normalized weighted mean). **b** Plots showing TCF7L2 (left panel) and MEF2C (right panel) associated genes. All known annotated targets from DOROTHEA are shown. Top10 absolute log2 fold change are labeled. Consistency between the log2 fold change and the prior knowledge (i.e., weight) is color coded. **c** Bar plot generated by Progeny analysis, showing the association with genes, which are deregulated in ZBTB18 FL-conditioned vs. EV-conditioned microglia. Color code represents the score from decoupleR enrichment analysis (normalized weighted mean). **d** Bodipy TMR-X lipid staining of microglia conditioned with EV or ZBTB18-FL expressing BTSC233 or BTSC475 cells (ctrl or upon ZBTB18 KO (sgZBTB18 #5(4)). Nuclei were counterstained with DAPI. Scale bar: 50 μm. Quantification of Bodipy TMR-X lipid staining shown in (**e**). n = 4 biological replicates; *P < 0.05, **P < 0.01.

observed microglia phenotypic changes are, at least in part, linked to the reduced secretion of those soluble factors. In particular, a reduced release of CCL2 would be in line with the previously reported role of CCL2 in GAM polarization to a tumor-promoting phenotype, in addition to its role as chemoattractant[60]. Of note, GDF-15 was recently shown to suppress macrophage-mediated immunosurveillance in pancreatic cancer and to promote oxidative metabolism and M2-polarization in adipose tissue macrophages[61]. The reduced expression of genes implicated in the oxidative phosphorylation is accompanied by a reduced OCR/ECAR ratio, consistent with previous reports that pro-tumorigenic microglia mostly rely on oxidative phosphorylation for its energetic supply[62]. Interestingly, microglia conditioned with ZBTB18 FL medium had a higher content of lipid droplets, which have been associated with inflammatory microglia[53], compared to microglia exposed to the medium of BTSC233 EV. An opposite result was observed in the ZBTB18 knock down experiment, corroborating the finding. Since oxidative phosphorylation also depends on fatty acid oxidation, the accumulation of lipid droplets could be the consequence of the reduced use of lipids for energy production[63]. Moreover, microglia conditioned with medium from ZBTB18 expressing cells showed a strong reduction of S100A4, a small calcium binding protein that is associated with poor prognosis in various cancers, and which has been linked to EMT and GBM mesenchymal transition[64,65]. Notably, S100A4 has been described as an important biomarker in GBM and appears to act upstream of master regulators of mesenchymal GBM[65]. Given the association of S100A4 with pro-tumorigenic GAMs[13] and the recognized role of GAMs in the establishment of mesenchymal phenotype[66], it is possible that S100A4 plays a major role in this process.

A further indication of a ZBTB18-mediated microglial inflammatory switch came from the observed increment of MHCII expression, both at the RNA and protein level, in microglia conditioned with the medium of ZBTB18 expressing cells. These results were corroborated in vivo, through staining of tumors derived from a syngeneic mouse model. However, it is important to highlight that the KAB203-C57BL/6 model significantly differs from the NOD/SCID one used in the rest of the manuscript, namely in the fact that it has a fully functional immune system. This might explain the observed differences in terms of survival and macrophages/microglia infiltration. It will be then crucial to expand the research on this model in order to validate these results and to increase the statistical power of the analyses. The full characterization of such a model will be key to understand how the tumor microenvironment is able to interact with a functional host immune system, which will be the object of our future research. Overall, each mouse model allows the characterization of different aspects of the microglia changes induced by ZBTB18 expression in GBM cells. In fact, while a strong effect on microglia migration can be observed in the BTSC233 model, the KAB203 model confirms in vivo the induction of MHCII, which was previously observed only in vitro and which is consistent with a change in the microglia phenotype. Supporting our results, previous studies have shown that the presence of T cells is important for MHCII expression in microglia[67] and this could explain why the same effect was not observed in the NOD/SCID mouse system.

The demonstration that ZBTB18 impairs the crosstalk between GBM cells and microglia reinforces the previously reported tumor suppressive role of ZBTB18 and its recent connection to metastases[16,29,68], strengthening its importance as potential target for therapy. Moreover, since we previously identified ZBTB18 as a negative regulator of the mesenchymal GBM[16,69], our results are consistent with other studies showing that the inactivation of negative regulators of the mesenchymal subtype (i.e., NF1) results in recruitment of GAMs[4].

Finally, our study confirms that ZBTB18 Nte-SF, which is generated by calpain-mediated cleavage[19], no longer exerts a tumor suppressive role. In fact, microglia cells exposed to GBM cells expressing ZBTB18 Nte-SF maintained a pro-tumorigenic phenotype, similar to what was observed upon exposure to BTSC-EV medium. However, RNAseq data showing the enrichment of a TNFα/NFκB associated signature suggest that ZBTB18 Nte-SF might be associated with an aberrant inflammatory process. Further

studies will be required to better understand the significance of these phenotypic changes and, in light of possible future therapeutic approaches aimed at re-expressing ZBTB18 in GBM cells, it would be important to completely suppress the expression of the cleaved product, to avoid any unexpected, adverse effect.

## Methods

### Ethical approval for human and animal subjects
Glioma biopsies were collected at the Department of Neurosurgery of the University Medical Center Freiburg (Freiburg, Germany) in accordance with an Institutional Review Board–approved protocol (Ethic Commission of Albert-Ludwigs University of Freiburg). Informed consent was obtained from all patients, in accordance with the declaration of Helsinki. All ethical regulations relevant to human research participants were followed.

The intracranial injection of BTSC233 and JX6 cells in NOD/SCID mice was previously reported[16]. The intracranial injection of KAB203 cells was performed in C57BL/6 mice according to the Directive 86/609/EEC of the European Parliament and of the Council of 24 November 1986; authorization for the animal research has been provided by the local authorities (Regierungspräsidium Freiburg). We have complied with all relevant ethical regulations for animal use.

### Cell culture
Patient-derived glioblastoma stem cells-like (BTSC233, BTSC268 and BTSC475) were generated in our laboratory at the University Clinic Freiburg in accordance with an Institutional Review Board–approved protocol (Ethic Commission of Albert-Ludwigs University of Freiburg)[16]. All ethical regulations relevant to human research participants were followed. The patient-derived JX6 GBM xenoline was previously described[16,70]. BTSCs were grown in Neurobasal medium (Life Technologies) containing B27 and N2 supplement (Life Technologies), FGF (20 ng/ml, R&D Systems), EGF (20 ng/ml, R&D Systems), LIF (20 ng/ml, Genaxxon Biosciences), Heparin (2 µg/ml, Sigma), and Glutamax (Invitrogen). GBM#22 sgLSD1#1 were previously described[20,29]. Human immortalized microglia (ABM-T0251; Applied Biological Materials), SNB19 and HEK293T cell lines were grown in 10% FBS DMEM. All cells were mycoplasma-free. SNB19 have been authenticated on 3/2/2017 and on 11/26/19 (SNB19 and HEK293T) by PCR-single-locus-technology (Eurofins Medigenomix). Data obtained in SNB19 cells were previously generated; however, in virtue of the listing of these cells in the database of commonly misidentified cell lines (ICLAC), we decided to validate the results using GBM patient-derived cells (BTSCs). To inhibit LSD1, BTSC475 were treated with the RN-1 inhibitor (Calbiochem, 5 µM) for 96 h.

### Lentiviral vectors and viral infection
The pCHMWS lentiviral vectors expressing FLAG-ZBTB18 FL and FLAG-ZBTB18 Nte-SF have been previously described[19,20]. To produce lentiviral particles, 293 T cells were transfected with the appropriate lentiviral vector along with pMD2G and psPAX2 packaging vectors. After 48 h, the supernatant containing the viral particles was incubated with 1 volume of Lenti-x concentrator (Takara Bio) every 3 volumes of medium, incubated overnight at 4 °C and centrifuged at 1500 g for 45 min at 4 °C. The pellet was then resuspended in 200 µl of sterile PBS, aliquoted and stored at −80 °C.

The sgZBTB18 #4 cloned in the pKLV-U6gRNA(BbsI)-PGKpur-o2BFP (Addgene, #50946) was previously described[20]. The sgZBTB18 #5 (5'-CTGAGCGAGCAGAGACACCA-3') was cloned in the pLentiCas9 vector (Addgene, #78546) at the BsmBI site. Briefly, the sgRNA#5-for (5'-CACCGCTGAGCGAGCAGAGACACCA-3') and sgRNA#5-rev (5'-AAACTGGTGTCTCTGCTCGCTCAGC-3') primers were annealed, BsmBI digested and ligated to the BsmBI-linearized pLentiCas9 vector. BTSC475 cells were transduced with lentiviral particles prepared as described in ref. 16. Cells were GFP sorted at the Lighthouse Core Facility of the University of Freiburg medical Center. Only the 20–30% strongest GFP positive cells were retained. Single clones were established by limiting dilution. *ZBTB18* knockout was verified by Sanger sequencing upon

amplification of the *ZBTB18* locus surrounding sgRNA#5. The following primers were used: sgRNA#5-seq-f (5'-TCCTCTCTCCCC AGGTTATG-3') and sgRNA#5-seq-r (5'- AGCAGCCACATAGCAGGC-3'). Sanger sequencing was performed with the sgRNA#5-seq-r primer. All primers were synthesized by Thermo Fisher Scientific. LSD1 silencing (shLSD1-71) was obtained by means of a pLKO lentiviral vector (TRCN0000046071, Sigma-Aldrich), as previously described[20,29].

### Chromatin Immunoprecipitation
Chromatin Immunoprecipitation (ChIP) was performed using the procedure previously described[20] 3 ug of the following antibodies were used: anti-HA (Abcam, #18181) and anti-ZBTB18 (Proteintech, #12714-1-AP). The immunoprecipitated DNA was analyzed by qRT- PCR using the following primers: CCL2_ChIP-for TGGTCAGTCTGGGCTTAATGG; CCL2_ChIP-rev TCAAGCAGGAGGAGGGATCTT; GDF15_ChIP-for CCGTTCTCCTCTGCTTCCTTT and GDF15_ChIP-rev GCCTCAG TATCCTCTTCCTCGA. Primers were synthesized by Thermo Fisher Scientific.

### RNA extraction and Real-Time qPCR
Total RNA was extracted from the cell culture using miRNeasy Mini Kit (QIAGEN) according to the manufacturer's instructions. First-strand cDNA synthesis was generated using the Superscript III First-Strand Synthesis System for RT–PCR (Life Technologies) following the manufacturer's protocol. Quantitative RT–PCR was performed using Kapa SYBR Fast (Sigma-Aldrich). The following primers (purchased at Thermo Fisher Scientific) were used: CCL2-for CAGCCAGATGCAATCAATGCC; C CL2-rev TGGAATCCTGAACCCACTTCT; GDF15-for GTGTTGC TGGTGCTCTCGTG; GDF15-rev CGGTGTTCGAATCTTCCCAG; CX 3CL1-for ACCACGGTGTGACGAAATG, CX3CL1-rev TGTTGATA GTGGATGAGCAAAGC; H2Aa-for CTGATTCTGGGGGTCC TCGC, H2Aa-rev CCTACGTGGTCGGCCTCAAT, H2Ab-for GAG CAAGATGTTGAGCGGCA, H2Ab-rev GCCTCGAGGTCCTTTCT GACTC, Hprt-for: TCAGTCAACGGGGGACATAA and Hprt-rev: GGGGCTGTACTGCTTAACCAG.

### Cytokine array
The cytokine content of culture medium conditioned by SNB19 or BTSC233 expressing empty vector (EV) control, full-length ZBTB18 (ZBTB18 FL), or N-terminal short form ZBTB18 (ZBTB18 Nte-SF) was assessed with a Human Cytokine Antibody Array C5 (R&D Systems) according to the manufacturer's instructions. In detail, cells were seeded in 10-cm plates at $1 \times 10^6$ cells/plate and infected with lentiviral vectors, according to the experimental plan. Then, 5 ml of the appropriate fresh medium (10%FBS DMEM for SNB19, Neurobasal medium containing B27 and N2 supplements for BTSC lines) were added to each plate and allowed to condition for 48 h. After this incubation, conditioned media were collected, centrifuged at 1000 rpm for 5 min to remove cell debris, and directly blotted on the cytokine arrays. Result analysis was performed according to the manufacturer's instructions: the spots blotted on the membranes were quantified using ImageLab 6.1 (Bio-Rad), selecting local background subtraction. The background calculated from the negative control spots was subtracted from the readings, which were then normalized by the average volume of the positive control spots of each membrane. Finally, individual spots from each experimental sample were normalized by the relative spot on the EV control membrane. In the case of SNB19, a cytokine array with unconditioned 10%FBS DMEM was run as an additional control, to exclude cytokines deriving from FBS.

### ELISA
GDF-15 and CCL2 concentration in culture medium conditioned GBM cell lines was quantified with a Quantikine ELISA Human GDF-15 (RayBiotech) or a Quantikine ELISA Human CCL2 (RayBiotech) according to the manufacturer's instructions.

### Immunoblotting
Total protein extracts were prepared as previously described[16]. Cytoplasmic and nuclear extracts were prepared using a Nuclear Extract Kit (Active-Motif, #40010) according to the manufacturer's instructions. Western blots were performed using the following antibodies: anti-FLAG (Sigma-Aldrich, #F1804, clone M2, 1:1000), anti-ZBTB18 (Abcam, #ab118471, 1:500); anti-LSD1 (Santa Cruz, #sc-53875, 1:1000); anti-H3K4me2 (Cell Signaling, #9725, 1:1000); anti-GDF15 (Sigma, #AMAb90687, 1:500); anti-alpha Tubulin (Abcam, #ab7291, clone DM1A, 1:5000), anti-HA (Abcam, #18181), clone HA.C5 (1:1000) and anti-Lamin A/C (Cell Signaling Technologies, #4777, clone 4C11,5, 1:2000). The ectopic expression of ZBTB18 in KAB203 cells was detected with an anti-ZBTB18 antibody (Proteintech, # 12714-1-AP, 1:1000). The quantification of western blot bands was performed using the Image Lab 6.1 software (Bio-Rad) and normalized to the level of αTubulin.

### Transwell invasion assay
The assay was conducted using transwell permeable invasion chambers (Costar) according to the manufacturer's instructions. Briefly, $2 \times 10^4$ microglia cells were seeded in the upper compartment. The conditioned medium from BTSC233 EV or BTSC233 ZBTB18 FL was used as chemoattractant, alone or in combination with CCL2 (20 ng/ml; R&D Systems), or GDF-15 (20 ng/ml; Peprotech). Alternatively, the conditioned medium from BTSC233 was used as chemoattractant, alone or in combination with 2.5 μg/ml of anti-CCL2 (Bio-Techne) or 2.5 μg/ml of anti GDF15 (Bio-Techne) neutralizing antibodies, according to the manufacturer's instructions. After 8 h, invading cells were fixed with 3% paraformaldehyde, 2% sucrose PBS and stained with 4',6-diamidino-2-phenylindole (DAPI; Sigma-Aldrich). Images were collected using a wide-field microscope (Axiovert, Zeiss).

### Intracranial injection
All experimental procedures were carried in accordance with the Directive 86/609/EEC of the European Parliament and of the Council of 24 November 1986; authorization for the animal research has been provided by the local authorities (Regierungspräsidium Freiburg). The experimental details regarding the intracranial injection of BTSC lines have been previously reported[16]. Briefly, six- week-old female NOD/SCID mice (Charles River Laboratories) were intracranially injected with $1.5 \times 10^5$ BTSC233 or JX6 cells (transduced with either a ZBTB18-expressing lentivirus or the empty vector). The animals were monitored and sacrificed when severe weight loss or neurologic symptoms occurred, based on our approved protocol. In none of the experiments were these limits exceeded.

Five-week-old C57BL/6 female mice (Charles River Laboratories) were intracranially injected (2 mm lateral and 0.5 mm anterior to the bregma, 3 mm below the skull) with $2 \times 10^5$ KAB203 mouse glioma cells (RCAS PDGFA/shpTrp53 mouse model[45]), either transduced with control (EV) or the ZBTB18-expressing pCHMWS lentiviral vector. Animals were monitored daily and sacrificed upon the development of neurological symptoms, according to the approved protocol. This limit was not exceeded. For histological analyses of mouse brain specimens, removed tissues were incubated overnight in 4% formalin, paraffin-embedded, and stained with H&E. Tumor size was estimated from the images of the H&E staining of tumor sections, using Adobe Photoshop CS5. For two brains per each group, microglia was extracted with CD11b beads (Miltenyl Biotech) according to the manufacturer's protocol.

### Immunostaining and lipid staining
Collected mouse brains were placed into 4% paraformaldehyde fixative overnight, and embedded in paraffin[16]. Sample sections were labeled with the following primary antibodies: anti-IBA1 rabbit monoclonal (ab178846; Abcam, clone EPR16588, 1:500), anti-ZBTB18 rabbit polyclonal (#12714-1-AP, Proteintech, 1:50), anti-F4/80 rat monoclonal (MA1-91124, Invitrogen, lone A3-1, 1:200) or anti-MHC Class II rabbit polyclonal (PA5-116876,

Invitrogen, 1:100). Cultured microglia were fixed in 3% paraformaldehyde and stained with anti-MHC Class II rabbit polyclonal (PA5-116876, Invitrogen, 1:100). Anti-rabbit IgG (H + L) Alexa Fluor 594 (1:200), anti-rabbit IgG (H + L) Alexa Fluor 488 (1:200) were used as secondary antibodies. Nuclei were counterstained with 4',6-diamidino-2-phenylindole (DAPI; Sigma-Aldrich). Stainings were analyzed for quantification with Adobe Photoshop CS5.

Lipid droplets were stained with 0.5 µg/ml Bodipy TMR-X SE (Thermo Fisher Scientific) in 150 mM NaCl for 10 min at room temperature. Nuclei were counterstained with 4',6-diamidino-2-phenylindole (DAPI, Sigma-Aldrich). All pictures were acquired using a FSL confocal microscope (Olympus). Stainings were analyzed with Adobe Photoshop CS5.

### Metabolic assay
Glycolysis and mitochondrial respiration rates of microglia cells were determined with a Seahorse ATP Production Rate Assay (Agilent) according to the manufacturer's instructions. Briefly, microglia was cultured for 48 h in 10% FBS DMEM or in DMEM conditioned by BTSC233 EV, BTSC233 ZBTB18 FL, or BTSC233 ZBTB18 Nte-SF. Subsequently, $7.5 \times 10^4$ microglia cells were seeded in each well of a XF96-well cell culture microplate in 200 µl of DMEM naive, or conditioned by BTSC233 EV, BTSC233 ZBTB18 FL, or BTSC233 ZBTB18 Nte-SF, respectively. Microglia was incubated overnight at 37 °C in 5% $CO_2$. The day of the experiment, the culture medium was replaced with 180 µl of 10 mM Glucose, 1 mM Sodium Pyruvate, 2 mM L-Glutamine Seahorse XF DMEM Medium, pH 7.4 (Agilent). The metabolic rates were measured by the sequential addition of 1.5 µM Oligomycin and 0.5 µM Rotenone/Antimycin A with a Seahorse XF96 Extracellular Flux Analyzer (Agilent).

### Gene expression profiling
Expression profiles of BTSC233 (GSE97347), JX6 (GSE97349) and SNB19 cells (GSE138890) included in this study were obtained from previously generated microarray datasets[16,20]. Transcriptome profiles of BTSC268 and BTSC475 cell lines, transduced with either EV or ZBTB18-FL, were generated by RNA-seq (GSE247079). To this end, total RNA was isolated as described above, and TruSeq library preparation and paired-end sequencing ($2 \times 100$ bp) on NovaSeq 6000 instrument (Illumina) was carried out at the Genomics and Proteomics Core Facility of the DKFZ, Heidelberg, using standard protocols. Each condition was assessed in triplicates. A detailed description of the analysis method is included in the Supplementary Materials.

### Long read RNA sequencing
Long read RNA sequencing was implemented to profile the transcriptomes from microglia, cultured for 48 h in 10% FBS DMEM or in DMEM conditioned medium obtained from BTSC233 EV, BTSC233 ZBTB18 FL, or BTSC233 ZBTB18 Nte-SF. To this end, total RNA was prepared as described above. mRNA purification from total RNA samples was performed using the Dynabeads mRNA Purification Kit (Thermo Fisher Scientific, Carlsbad, USA). For reverse transcription reaction the SuperScript IV reverse transcriptase (Thermo Fisher Scientific, Carlsbad, USA) was used. Libraries were prepared by using the Low Input by PCR Barcoding Kit and the cDNA-PCR Sequencing Kit (Oxford Nanopore Technologies, Oxford, United Kingdom) as recommended by the manufacturer.

RNA sequencing was performed using the MinION Sequencing Device, the SpotON Flow Cell and MinKNOW software (Oxford Nanopore Technologies, Oxford, United Kingdom) according to the manufacturer's instructions. Samples were sequenced for 48 h on two flow-cells. Basecalling was performed by Albacore implemented in the nanopore software. Genes that were quantified (>0 counts) in less than 50% of the dataset were excluded from further downstream analyses. Library size and TMM normalization was applied to raw count matrices. GEO accession number: GSE227722.

### Gene correlation and survival analysis
ZBTB18, AIF1/IBA1, CCL2, GDF15, CXCL8, and ANG gene correlation, gene expression, and patient survival analyses were performed using the GlioVis[23] and cBioPortal[24] platforms, and data from The Cancer Genome Atlas (TCGA) (https://www.cancer.gov/tcga), using the default settings. Briefly, for CCL2, GDF15, and ANG expression and correlation with ZBTB18 in GlioVis, we mined the TCGA_GBMLGG database; while Kaplan-Meyer curves were calculated on the TCGA_GBM database. We calculated Pearson's correlation, including all histologies and all subtypes. It must be mentioned that in GlioVis ZBTB18 is still called with his old name (ZNF238) in the TCGA database. For cBioPortal analyses, GBM (TCGA, PanCancer Atlas) database was mined for mRNA expression in all samples.

### Statistics and reproducibility
For the statistical analysis of the RT-qPCR data are considered to be statistically significant when $p < 0.05$ by two-tailed Student's $t$ test. The number of replicates and the definition of biological versus technical replicates is indicated in each figure legends. All graphs and statistical analyses were generated using GraphPad Prism 9.

### Importance of the Study
Several studies have highlighted the role of the microenvironment in supporting tumor growth. In particular, resident microglia and macrophages recruited from circulation, release cytokines that promote tumor progression and favor the establishment of an immunosuppressive milieu. This activity is strongly supported by the tumor cells, which produce and release a variety of factors, crucial for the tumor-promoting function of microglia/macrophages. The regulatory feedback loop between the tumor cells and these macrophage populations ultimately drives the establishment of the most aggressive tumor traits and prevents the mounting of an effective immune response. ZBTB18 is a negative master regulator of the mesenchymal phenotype in glioblastoma, and its newly described role as a transcriptional repressor of key cytokines suggests a way to disrupt the aberrant regulatory loop between tumor and immune cells. Thus, methods to re-express ZBTB18 in glioblastoma cells could lead to important advancement in immunotherapy against this disease.

### Reporting summary
Further information on research design is available in the Nature Portfolio Reporting Summary linked to this article.

### Data availability
Microglia and BTSC RNA-seq data have been deposited at GEO and are publicly available as of the date of publication. GEO accession numbers: GSE227722 (microglia) and GSE247079 (BTSC268 and BTSC475). Gene expression array in BTSC233 (GSE97347), JX6 (GSE97349) and SNB19 cells (GSE138890) have been previously described[16,20]. All source data are included in Supplementary Data 1. All uncropped gel pictures are displayed in Supplementary Fig. 13-14. All other data are available from the corresponding author on reasonable request.

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

## Acknowledgements
We thank Mohammad Al Shaab, Sofia Munari, Andrea Gatta, and Jonathan Goldner for technical assistance and Roberto Accolla and Greta Forlani for critically reading the manuscript. We are grateful to Oliver Gorka and Olaf Gross for support with Seahorse analysis. The results shown here are in part based upon data generated by the TCGA Research Network: https://www.cancer.gov/tcga. This work was partially financed by the German's Cancer Aid grant (Deutsche Krebshilfe, 70113120), Müller-Fahnenberg-Foundation and Research Committee (Forschungskomission) of the Faculty of Medicine, University of Freiburg.to MSC. We acknowledge funding from the Deutsche Forschungsgemeinschaft (DFG) within the CRC1160 (Project ID 256073931-Z02 to M.B.), CRC/TRR167 (Project ID 259373024-Z01, M.B.), CRC1453 (Project ID 431984000-S1, M.B.), CRC1479 (Project ID: 441891347-S1 to M.B.). We also acknowledge funding from the German Federal Ministry of Education and Research (BMBF) within the Medical Informatics Funding Scheme-MIRACUM-FKZ 01ZZ1801B (M.B.) and PM4Onco-FKZ 01ZZ2322A (M.B.) and EkoEstMed–FKZ 01ZZ2015 (G.A.). M.S. is supported by the Berta-Ottenstein-Programme for Clinician Scientists and the IMMPACT-Programme for Clinician Scientists, Department of Medicine II, Medical Center, University of Freiburg and Faculty of Medicine, University of Freiburg, funded by the Deutsche Forschungsgemeinschaft (DFG, German Research Foundation; 413517907). The Lighthouse Core Facility is funded in part by the Medical Faculty, University of Freiburg (Project Numbers 2021/A2-Fol; 2021/B3-Fol) and the DFG (Project Number 450392965).

## Author contributions
R.F., I.V.H., F.Z., E.K., and A.I. performed experiments. K.J., G.A., and E.C. performed RNA-seq analysis. M.S. and M.P. provided reagents and support for histology analysis. M.B. provided support for BTSC RNAseq and RNAseq analysis. D.H.H. and V.M.R. provided support for microglia RNAseq and stRNA. R.F. and M.S.C. conceived and supervised the study. R.F. and M.S.C. wrote the manuscripts, with input and editing support from all authors.

## Funding

## Competing interests
The authors declare no competing interests.
