## [Transparent Peer Review file · Communications Biology]

ZBTB18 regulates cytokine expression and affects microglia/macrophage recruitment and commitment in glioblastoma

Corresponding Author: Dr Maria Stella Carro

A version of this paper was originally rejected for publication by Communications Biology, however that decision was reconsidered after appeal by the authors.

Version 0:

Reviewer comments:

Reviewer #1

(Remarks to the Author)

In this manuscript Ferrarese et al. found that overexpression of ZBTB18 in glioblastoma cells impairs the production of specific cytokines, which have previously been associated with the recruitment of monocyte, macrophages, and microglia to the tumor microenvironment. They also show that the conditioned medium from glioblastoma cells that overexpress ZBTB18 have a lower ability to chemoattract human immortalized microglia in an in vitro transwell invasion assay. Furthermore, though RNAseq analyses done on microglia cells treated with conditioned medium from glioma cells with or without overexpression of ZBTB18, they show that the ZBTB18 status in tumor cells alters the phenotype of microglia in a manner that suggests the loss of the immune-suppressive phenotype and the acquisition of pro-inflammatory features by microglia.

ZBTB18 is a transcriptional repressor that is known to play a tumor suppressive role in glioblastoma and this study contributes to better understanding its possible role in cancer progression. The manuscript is well written, the experiments are overall well performed, and the discussion is very extensive. However, there are also weaknesses. Specifically, the findings presented in this manuscript are solely correlative and no direct functional assessment of whether ZBTB18 is necessary and/or sufficient for the recruitment of macrophage/microglia to tumors in vivo is provided. Furthermore, it appears that low levels of ZBTB18 do not strictly correlate with high levels of monocyte/microglia in patients, suggesting that additional factors are necessary for the recruitment of monocyte/microglia in glioblastoma.

Major points:

The title would need to be changed. The current title overstates the findings presented in the manuscript. There is currently no functional data in the manuscript to directly demonstrate that "ZBTB18 hinders...microglia and macrophage recruitment" and that "ZBTB18 hinders... the establishment of a tumor-supportive microenvironment in glioblastoma".

Fig.1. Please provide Western blot data confirming the ZBTB18 expression levels in the cell lines used to generate Fig. 1. Also, considering the presence of a short and long form of ZBTB18 in some glioblastoma cell lines, defining whether overexpression of ZBTB18 results exclusively in full length ZBTB18 or whether this is (in part) processed to short form would be important for data interpretation.

Fig. 3. Could the authors provide experimental details to describe the exact experiment performed in Fig. 3A and B? Details such as how the tumors were inoculated and when/at what stage the animals were sacrificed should be provided. Also, it would be important to provide the tumor mass/size for each mouse and the expression levels of ZBTB18 in the tumors. With the information provided currently, it is difficult to assess whether the changes in microglia/macrophages are directly correlating with changes ZBTB18 levels or whether differences in tumor size/stage also contribute to the phenotype.

Fig 3C-D and Sup. Fig.3C. The data in Fig. S3C suggests that many tumors display low levels of both CCL2 and ZBTB18. Similarly, in Fig. 3D, many tumors have low levels of both ZBTB18 and the microglia/macrophage marker. This suggest that low ZBTB18 expression is not sufficient to induce CCL2 and microglia/macrophage recruitment into tumors. Could the authors discuss how these data fit with the model they propose, in the context of tumor progression?

Fig. 3F. Could the authors deplete or inhibit CCL2 and GDF15 in the conditioned medium from EV cells? This would be necessary to demonstrate that these proteins are indeed responsible for the recruitment of microglia by the tumor cell conditioned medium in vitro. Currently, the data shows that CCL2 and GDF15 can recruit microglia (irrespective of the composition of the conditioned medium) but not that they are indeed the factors involved in microglia recruitment by the tumor cell conditioned medium.

Figure S2. Could the authors show a correlation of ZBTB18 expression with tumor grade, similar to what is currently shown for CCL2, GDF15, and ANG?

Figure S1B and C. Please provide quantitation. In particular, the ZBTB18 Nte-SF sample appears to have a higher background than the two others, so it is difficult to assess visually whether CCL2 levels are similar to those of the EV sample or not. Also, please provide details on how the conditioned media were prepared and how the samples were normalized.

The authors note a few discrepancies between the effect of ZBTB18 on cytokine expression at the mRNA and protein levels. In addition, they note: "A negative correlation was also observed between ZBTB18 and ANG, which seems to contradict the outcome of the cytokine array." Could the authors provide a possible explanation for these contradictory results?

Please provide more details on which data sets were mined and how the correlations were defined in GlioVis and cBioPortal. It appears that the ZBTB18 gene is not included in the TCGA_GBMLGG dataset in GlioVis. Please explain how the analyses presented in Fig. 3E, which include analyses of the correlation of ZBTB18 with AIF1 in the TCGA_GBMLGG dataset were done. Was a different data set mined?

Minor point:

Please provide a rationale for why some experiments are done with some cell lines and others are done with different cell lines. This would help strengthen the logical flow of the paper.

Reviewer #2

(Remarks to the Author)

The manuscript analyzed expression of ZBTB18 in different GBM samples and in overexpression/knockdown GBM cell lines. They showed that ZBTB18 affect the expression of a panel of cytokines through potential epigenetic regulation. ZBTB18 promotes glioblastoma progression has been reported by the same group a few years ago. In this manuscript, the authors mainly focused on the regulation of CCL2 and other cytokines by ZBTB18, and showed overexpression of ZBTB18 can increase the infiltration of microglia in mouse tumor xenograft.

1. In Fig1a, the backgrounds of the 5 GBM cell lines are not clearly stated. Please specify which cell line is with or without ZBTB18 overexpression. I also would like to see the western blotting data to show the base line and overexpression of ZBTB18 in all the cell lines used in this figure.

2. The Fig1c did not have CX3CL1 labeling.

3. The most reliable assay to check the binding activity of ZBTB18 on transcript promoter is to use comparison between ZBTB18 protein (ZBTB18 Abs pulled) vs IgG or Input control in Fig 1F.

4. The calculation in the table of Fig.1I seems to be wrong.

5. There is no quantitatively analyzing of correlation between ZBTB18 and GDF15/CCL2.

6. Based on the plot provided on 2E, one cannot conclude the negative association between ZBTB18 and GDF15/CCL2. Meanwhile, it could be helpful if the authors could label the infiltrating area or tumor core on the images represented.

7. "Taken together, these data suggest that ZBTB18 exerts a repressive action on the expression and consequent release of cytokines that could play a relevant role in shaping the tumor microenvironment and ultimately, influence the patients' clinical outcome."

The data presented in the manuscript are not compelling enough to conclude the above statement.

8. SupFig7A, please provide images of Iba1 and F4/80 co-staining.

9. In Fig3A and SugFig7C, why does the intensity of Iba1 decreased in the ZBTB18 overexpression samples? Could the authors provide another microglia marker IF image besides Iba1? And in fig3A, what's the difference between the left and right panel? In Fig.3A, left lower panel, we still could detect a similar amount of IBA+ cells as upper panel indicating BTSC233 EV group. The obvious difference was less intensity of IBA1 florescence. Please explain this difference.

10. Statistics is needed in Fig.3D.

11. In Fig4B, it is not convincing that ZBTB18 FL-conditioned microglia grouped separately from the ZBTB18 Nte-SF- and EV-exposed samples, which instead clustered together.

12. More detailed information is needed for describing for analyzing DEGs in Fig.6B.

13. For BTSC475 ZBTB18 KO cell lines, the knockout was confirmed by sanger sequencing, but some normal size ZBTB18 protein are still detected by western. Are these cells a mixed population with wildtype ZBTB18?

14. To compare the infiltration of microglia/macrophage cells in vivo, only ZBTB18 overexpression GBM line is used. I would like to see whether BTSC475 ZBTB18 KO has more infiltrated microglia than BTSC475 ctrl in the xenograft as well.

15. For the transwell invasion assay, could the authors provide the images of migrated microglia cells besides the statistics analysis? I also would like the authors to add the panel of microglia cells with EV conditioned medium and blocking antibodies of either CCL2 or GDF-15 or both.

16. While knocking out ZBTB18 increase CCL2 expression and doesn't affect GDF-15 expression. Is the recruitment of CTBP2 and LSD1 at CCL2 and GDF-15 promoters affected? Do the H3K4me and H3K9me change accordingly?

17. Does knockout ZBTB18 in GBM cells affect the mice survival or tumor growth after intracranial injection?
18. Fig1F Figure legend should include the full-length ZBTB18 overexpression.
19. Line 957, redundant "the"
20. I would like to see the authors to discuss the contradicting results of ANG in discussion.
21. In the Fig1I, the expression changes of CCL2 is 0.69 while IL8 is 34.18. This is a reversed trend to the main text. I'm not sure this is the correct calculation by dividing BTSC475-sgZBTB18 #5 by BTSC475-ctrl.

Reviewer #3

(Remarks to the Author)

This manuscript highlights the role of ZBTB18, derived from GBM tumor cells, in regulating expression of TAM chemoattractant CCL2 and GDF15, and in inducing pro-inflammatory phenotypes of microglia, supporting that ZBTB18 overexpression in GBM cells may have therapeutic efficacy in treating GBM. While the regulatory role of GBM-expressing ZBTB18 on TAM recruitment and phenotypical switch is interesting, this manuscript lacks solid experimental evidence to prove the mechanistic hypothesis (mostly based on in silico bioinformatics), which raised serious major concerns that must be addressed as follows:

1. ZBTB18 Nte-SF, a derived product of ZBTB18, is used as another control group in most of the experiments. The results suggested that this byproduct Nte-SF, played an opposite role from ZBTB18. The authors may want to explain (1) why they used this Nte-SF-OV as a independent group, (2) how this byproduct is produced from ZBTB18, (3) the likelihood of introducing this tumor-promotive byproduct into GBM cells while applying ZBTB18 overexpression therapeutic approach, (4) to what extent this byproduct may disrupt the tumor-repressive effect of ZBTB18.
2. The authors constantly used gain-of-function(ZBTB18 overexpression) experiments to demonstrate the downstream functions (e.g., the negative regulatory effect of ZBTB18 on chemoattractant CCL2 and GDF15), while loss-of-function experiments were rather lacking throughout the manuscript. ZBTB18 KO even exerts no change on GDF-15. These raised serious concern about the potency of regulatory effect from ZBTB18 on the proposed downstream functions.
3. The authors indicate the topic in the title as glioblastoma; however, when proving clinical relevance of ZBTB18, the analyzed datasets included LGG as well (in Figure S2B, Figure 3E, etc.). Will the significance be impaired if datasets only containing GBM samples were analyzed?
4. The mechanistic study of ZBTB18 transcriptionally regulating CCL2 and GDF15 is not sufficient. (1) Line 306: EMSA is suggested to prove direct binding of ZBTB18 at the CCL2 and GDF15 promoter. (2) What promotes ZBTB18 to be capable of directly regulating several chemokines? (3) The authors may also design rescue experiments to demonstrate that ZBTB18 transcriptionally represses CCL2 and GDF15 via dysregulating histone demethylase activity.

Other concerns:

1. The correlation between ZBTB18 and its downstream effectors, as revealed by bioinformatics in Figure S2B, Figure 3E, should be validated experimentally.
2. Line 333: how did the authors draw the conclusion that ZBTB18 expression is localized to GBM cells as no other non-GBM cells was displayed in Figure 2A.
3. Line 385: no show of ZBTB18 Nte-SF group in Figure 3F.
4. Line 408/Figure 4B: possible overstatement. No clear discrepancy among the red, blue and green dots (ctrl, EV, ZBTB18 FL).
5. Figure 5H: not convincing at all. What is the situation under in vivo conditions?

Version 1:

Reviewer comments:

Reviewer #1

(Remarks to the Author)

The authors have satisfactorily addressed many of the reviewers' comments and added new data and clarifications that strengthen the manuscript. However, some concerns remain.

Specific points:

1) As mentioned in the first round of review, "it would be important to provide the tumor mass/size for each mouse and the expression levels of ZBTB18 in the tumors. With the information provided currently, it is difficult to assess whether the changes in microglia/macrophages are directly correlating with changes ZBTB18 levels or whether differences in tumor size/stage also contribute to the phenotype".

These points have not been convincingly addressed by the authors. They indicated that tumors have comparable sizes and referred to Fedele et al., 2017 Fig.4C-D for data showing this. However, there is no measurement of tumor size in those figures. Could the authors provide a quantitation? It would also be relevant to confirm the expression levels of ZBTB18 in those tumors.

2) Line 339-342: The discussion of supplementary figure 3 should be modified. Although fitting curves on this data suggests

a negative correlation, the biological relevance of this conclusion is questionable. Indeed, as shown previously by the authors, the vast majority of GBM express very low levels of ZBTB18 (if any). Thus, any correlation between ZBTB18 expression levels and that of other markers is very difficult to make. The conclusion here should be that various levels of CCL2 are expressed by GBM, which have low levels of ZBTB18 expression.

3) Figure 2. The effect of ZBTB18 knockout on CCL2 protein expression level is not convincing. Specifically, while a slight increase is seen in Fig 2a and 2b (with clone 24), no difference is seen in Fig 2b (with clone 4) and 2f, despite these experiments all being a very similar. This should be better explained in the summary of this section on lines 400-403.

Furthermore, the confirmation of knockout for clone 24 should be shown.

4) Fig. 3d. A scale bar for the meaning of the different colors could be helpful.

5) Fig 3e. The data in Fig.3e show a lack of correlation between ZBTB18 expression and the other markers (values are close to 0). Inverse correlations would show value close to -1. Therefore, the authors should change the wording of their conclusion (lines 410-413) to indicate a lack of correlation.

6) Supplementary figure 2b: Please add statistics.

Reviewer #2

(Remarks to the Author)

The authors have addressed all my concerns.

Reviewer #3

(Remarks to the Author)

These concerns remain to be addressed:

1. For major concern 1: the authors admitted that the two forms of ZBTB18 (with opposite functions) co-exist, but they did not indicate where are the supporting data for "In terms of therapeutic approach, we have shown that the co-existence of the cleaved product does not impair the repressive and tumor suppressive function of ZBTB18 FL,..."
2. For major concern 3: the authors did not answer my questions. What would the correlation be like in Fig. S2b and the revised Fig. 4e if the TCGA_GBM dataset rather than TCGA_GBMLGG was applied for analysis? Will it still be significant?
3. For major concern 4: (1) missing words in the sentence of the second paragraph? ("the presence of a motif does not necessarily means that the TF can bind and the.") (2) The authors did not perform the suggested experiments or did not show the results in the point-by-point response letter which they decided not to include in the revised manuscript.
4. For other concern 5: low quality and unconvincing images presented in Figure 5H. No more convincing data (e.g. flow cytometry, IHC staining of slides from in vivo models) were included in the revision.

Version 2:

Reviewer comments:

Reviewer #1

(Remarks to the Author)

The authors have adequately responded to my comments. However, I have concerns with the newly added Figure 7 and Supplementary Figure 12. Specifically:

- 1) Sup. Fig. 12a: A loading control is missing. Please include a reprobing of the same membrane with anti-Tubulin.
- 2) Sup. Fig. 12b, c: Why were only 3 EV tumors included in the analyses? The authors mentioned that 7 out of 9 EV tumors grew and that "one tumor in the EV group had cells growing both in the striatum and outside the cortex, and therefore was not included in the subsequent examinations". Thus 6 tumors should remain and be included in subsequent analyses presented in Figure 7d, e and Supplementary Figure 12 b, c, d, g, h.

Furthermore, while the manuscript mentions that only 3 out of 9 tumors grew in the ZBTB18 FL group, the rebuttal letter mentions that 6 out of 9 tumors from this group grew. Please make sure the correct information is reported in the manuscript.

3) Sup. Fig. 12f: Please better explain where the meninges are and where interaction with the tumor cells can be seen. A higher magnification image and/or an arrow might be helpful.

4) In the BTSC233 model (in NSG mice), ZBTB18 suppresses tumor growth, but these tumors do not express MHCII. In contrast, the new data with the KAB203 model (in immunocompetent mice) shows that ZBTB18 increases MHCII expression, but it remains unclear whether ZBTB18 affects tumor growth and survival in this model. Taken together, data from these two models argue against MHCII playing an important role in the anti-tumorigenic effect of ZBTB18.

How can these results be reconciled with the central message of the manuscript, which proposes that ZBTB18 regulate microglia-tumor cell interactions and that this presumably affects tumor growth?

Reviewer #2

(Remarks to the Author)
I have no further comments.

Reviewer #3

(Remarks to the Author)
The authors have addressed most of my concerns.

Version 3:

Reviewer comments:

Reviewer #1

(Remarks to the Author)
The authors have satisfactorily addressed most of the comments.

However, issues with the data in Fig. 7 and Sup. Fig. 12 remain.

- In their rebuttal the authors wrote (regarding the model used in Fig. 7 and Sup. Fig. 12): “however, in some cases, tumor cells grew in the subarachnoid space outside the meninges causing a premature death of these animals. Thus, it is not possible to accurately determine the survival rate in the two conditions.” This comment raises questions about Sup. Fig. 12e, which directly compares the survival rate of these two conditions. Considering their comments, the authors should remove Sup. Fig. 12e from the manuscript. In addition, they should better explain the limitations of their model in the text (similar to what they did in the rebuttal).

- The q-RT PCR analysis in Figure 7f has only one biological replicate per group (n=1). If no additional biological replicates are available, please consider removing this panel. Alternatively, clearly indicate this limitation in the main text of the manuscript.

Reviewers' comments:

Reviewer #1 (Remarks to the Author):

In this manuscript Ferrarese et al. found that overexpression of ZBTB18 in glioblastoma cells impairs the production of specific cytokines, which have previously been associated with the recruitment of monocyte, macrophages, and microglia to the tumor microenvironment. They also show that the conditioned medium from glioblastoma cells that overexpress ZBTB18 have a lower ability to chemoattract human immortalized microglia in an in vitro transwell invasion assay. Furthermore, though RNAseq analyses done on microglia cells treated with conditioned medium from glioma cells with or without overexpression of ZBTB18, they show that the ZBTB18 status in tumor cells alters the phenotype of microglia in a manner that suggests the loss of the immune-suppressive phenotype and the acquisition of pro-inflammatory features by microglia. ZBTB18 is a transcriptional repressor that is known to play a tumor suppressive role in glioblastoma and this study contributes to better understanding its possible role in cancer progression. The manuscript is well written, the experiments are overall well performed, and the discussion is very extensive. However, there are also weaknesses. Specifically, the findings presented in this manuscript are solely correlative and no direct functional assessment of whether ZBTB18 is necessary and/or sufficient for the recruitment of macrophage/microglia to tumors in vivo is provided. Furthermore, it appears that low levels of ZBTB18 do not strictly correlate with high levels of monocyte/microglia in patients, suggesting that additional factors are necessary for the recruitment of monocyte/microglia in glioblastoma.

We thank the Reviewer for his/her comments. We have addressed his/her concerns below.

Major points:

The title would need to be changed. The current title overstates the findings presented in the manuscript. There is currently no functional data in the manuscript to directly demonstrate that “ZBTB18 hinders...microglia and macrophage recruitment” and that “ZBTB18 hinders... the establishment of a tumor-supportive microenvironment in glioblastoma”.

As suggested by the reviewer, we propose to change the title to: **“ZBTB18 regulates cytokine expression and affects microglia/macrophage recruitment and commitment in glioblastoma”**

Fig.1. Please provide Western blot data confirming the ZBTB18 expression levels in the cell lines used to generate Fig. 1. Also, considering the presence of a short and long form of ZBTB18 in some glioblastoma cell lines, defining whether overexpression of ZBTB18 results exclusively in full length ZBTB18 or whether this is (in part) processed to short form would be important for data interpretation.

The western blot data for BTSC233, JX6 and SNB19 have been shown in our previous studies (Fedele et al., 2017 Fig. 2C and 2E; Ferrarese et al., 2023, Fig.2A and 2G; Masilamani et al., 2022, Fig. 1A, Fig.5A). The figure below shows the western blots (with FLAG or ZBTB18-directed antibodies) to confirm the expression of ZBTB18 in BTSC268 and BTSC475 (new experiments). We have included this data in the revised version of the manuscript (Supplementary Figure 1a-b). In all the experiments, the overexpression of the full-length protein is accompanied by the processed short form; however, it has been shown in our

previous publications (Ferrarese et al., 2023; Masilamani et al., 2022) that, compared to the full length, short form ZBTB18 has different properties, only detectable when it is ectopically expressed in GBM cells and not when generated as by-product of the full length overexpression. In fact, ZBTB18 SF, which is localized in the cytoplasm, does not interfere with ZBTB18 FL direct gene repression function in the nucleus.

Figure 1. (A) Western blot analysis of BTSC475 and BTSC268 cells upon ectopic expression of ZBTB18 FL. ZBTB18 FL was detected with anti-FLAG M2 antibody (Sigma, left panel) or anti-DYKDDDDK (FLAG, Cell Signaling, right panel). (B) Western blot analysis of BTSC475 and BTSC268 cells upon ectopic expression of ZBTB18 FL. ZBTB18 FL was detected with anti-ZBTB18 antibody (Abcam, low and high exposure).

Fig. 3. Could the authors provide experimental details to describe the exact experiment performed in Fig. 3A and B? Details such as how the tumors were inoculated and when/at what stage the animals were sacrificed should be provided. Also, it would be important to provide the tumor mass/size for each mouse and the expression levels of ZBTB18 in the tumors. With the information provided currently, it is difficult to assess whether the changes in microglia/macrophages are directly correlating with changes ZBTB18 levels or whether differences in tumor size/stage also contribute to the phenotype.

The experimental details regarding the intracranial injection of BTSC233 cells have been previously reported (Fedele et al., 2017). Here is the information: six-week-old NOD/SCID mice (Charles River Laboratories) were intracranially injected with 1.5×10^5 BTSC233 cells

(transduced with either a ZBTB18-expressing lentivirus or the empty vector). The animals were monitored and sacrificed when symptoms occurred, based on our approved protocol. We have included this information in the revised manuscript. Regarding the tumor size, the tumors detected in the ZBTB18 group had a comparable size with respect to the empty vector group (Fedele et al., 2017 Fig.4C-D); however, tumors in the ZBTB18 group developed later with a consequent improved survival of the mice. Therefore, we suggest that the changes in microglia/macrophages affected the rate of tumor development. We have included this comment in the revised version of the manuscript.

Fig 3C-D and Sup. Fig.3C. The data in Fig. S3C suggests that many tumors display low levels of both CCL2 and ZBTB18. Similarly, in Fig. 3D, many tumors have low levels of both ZBTB18 and the microglia/macrophage marker. This suggest that low ZBTB18 expression is not sufficient to induce CCL2 and microglia/macrophage recruitment into tumors. Could the authors discuss how these data fit with the model they propose, in the context of tumor progression?

It has to be noted that in our analysis shown in former Figure 3C-D (now Figure 4c-d), most of the tumors with low IBA positivity are low grade gliomas, where it is known TAM infiltration is not strong. In Supplementary Figure 3c, cBioportal *in silico* analysis highlights mutual exclusivity between *CCL2* and *ZBTB18* within the TCGA, PanCancer Atlas dataset, albeit it seems that in most of the samples, neither *CCL2* nor *ZBTB18* are expressed. This suggests that *CCL2* is not constitutively expressed in gliomas, but most likely, it is a cell-type- and context-dependent activation. However, our data suggest that ZBTB18 is a repressor of CCL2 and, even though it might not be the only factor responsible for the modulation of this cytokine, the mutual exclusivity emerging from the analysis shown in Figure S3c confirms the fact that *CCL2* is not expressed in the presence of ZBTB18. FigureS2a-b shows that *CCL2* expression is higher in GBM compared to LGG and that it is inversely correlated to *ZBTB18* expression; this similarly suggests that *CCL2* is expressed when ZBTB18 is absent. Regarding IBA1 staining in former Figure 3C-D (now Figure 4c-d), the range of TAM marker positivity within samples with low ZBTB18 expression is indeed fairly wide, but also in this experiment, none of the samples with high ZBTB18 expression showed high degree of TAM infiltration, confirming the role of ZBTB18 as negative regulator of TAM recruitment. In conclusion, our analyses hint that ZBTB18 presence is sufficient to block CCL2 expression (and TAM recruitment), but it seems not to be the only factor able to keep the synthesis and secretion of this cytokine at bay. We agree with the Reviewer that a more complex regulation of cytokine expression is taking place in glioma cells and we have stated it more clearly in the manuscript text.

Fig. 3F. Could the authors deplete or inhibit CCL2 and GDF15 in the conditioned medium from EV cells? This would be necessary to demonstrate that these proteins are indeed responsible for the recruitment of microglia by the tumor cell conditioned medium *in vitro*. Currently, the data shows that CCL2 and GDF15 can recruit microglia (irrespective of the composition of the conditioned medium) but not that they are indeed the factors involved in microglia recruitment by the tumor cell conditioned medium.

We agree with the Reviewer and we have performed the proposed experiment (new Figure 4g). The data indeed indicate that blocking CCL2 with a specific antibody impairs microglia invasion while neutralizing GDF15 does not produce the same effect. Therefore, we conclude that ZBTB18 impairs microglia invasion mainly through CCL2 repression.

Figure 2. Invasion assay of microglia exposed to culture medium of BTSC233 cells treated with CCL2 or GDF-15 neutralizing antibodies. n=3 biological replicates; error bars \pm s.d.; *P < 0.05, **P < 0.01, and ***P < 0.001 by a t test.

Figure S2. Could the authors show a correlation of ZBTB18 expression with tumor grade, similar to what is currently shown for CCL2, GDF15, and ANG?

The correlation of *ZBTB18* expression with tumor grade was previously reported (Fedele et al.2017). *ZBTB18* is more expressed in low grade tumors. Below is an updated panel generated with GlioVis.

Figure 3. GlioVis analysis of *ZBTB18* expression in different-grade gliomas from the TCGA_GBMLGG dataset.

Figure S1B and C. Please provide quantitation. In particular, the ZBTB18 Nte-SF sample appears to have a higher background than the two others, so it is difficult to assess visually whether CCL2 levels are similar to those of the EV sample or not. Also, please provide details on how the conditioned media were prepared and how the samples were normalized.

We have included a quantification of CCL2 spot on the cytokine arrays, normalized to the relative controls (Supplementary Fig. 1f). We are also including the table here:

Sample		CCL2
SNB19	EV	1
	ZBTB18 FL	0,11
BTSC233	EV	1
	ZBTB18 FL	0,18
	ZBTB18 Nte-SF	1,08

As for the methods, cells were seeded in 10-cm plates at 1×10^6 cells/plate and infected with lentiviral vectors, according to the experimental plan. Then, 5 ml of the appropriate fresh medium (10%FBS DMEM for SNB19, Neurobasal medium containing B27 and N2 supplements for BTSC lines) were added to each plate and allowed to condition for 48 hours. After this incubation, conditioned media were collected, centrifuged at 1000 rpm for 5 minutes to remove cell debris, and directly blotted on the cytokine arrays. Result analysis was performed according to the manufacturer's instructions. Briefly, the spots blotted on the membranes were quantified using ImageLab 6.1 (Bio-Rad), selecting local background subtraction. The background calculated from the negative control spots was subtracted from the readings, which were then normalized by the average volume of the positive control spots of each membrane. Finally, individual spots from each experimental sample were normalized by the relative spot on the EV control membrane. In the case of SNB19, a cytokine array with unconditioned 10%FBS DMEM was run as an additional control, to exclude cytokines deriving from FBS. We have included this description in the method section (Cytokine array).

The authors note a few discrepancies between the effect of ZBTB18 on cytokine expression at the mRNA and protein levels. In addition, they note: "A negative correlation was also observed between ZBTB18 and ANG, which seems to contradict the outcome of the cytokine array." Could the authors provide a possible explanation for these contradictory results?

We do not have a clear explanation for the ANG contradictory result. ZBTB18 overexpression does not induce *ANG* expression as indicated by microarray and RNAseq (Figure 1a and Supplementary Figure 1c). The correlation analysis based on mRNA expression suggests that *ANG* expression is inversely correlated to *ZBTB18* but not that ZBTB18 exerts a regulatory role on *ANG*. However, given the confirmation of a change in ANG secretion upon ZBTB18 overexpression and silencing, it is possible that ZBTB18 in some ways affects the protein release in the extracellular space. The same seems to occur for GDF15, according to the new data shown in Fig.2. We do not know whether these factors are post-translationally modified and/or differentially processed upon *ZBTB18* expression. To answer this question would require further studies that go beyond the scope of this manuscript. We have discussed these possibilities in the revised version of the manuscript.

Please provide more details on which data sets were mined and how the correlations were defined in GlioVis and cBioPortal. It appears that the ZBTB18 gene is not included in the TCGA_GBMLGG dataset in GlioVis. Please explain how the analyses presented in Fig. 3E, which include analyses of the correlation of ZBTB18 with AIF1 in the TCGA_GBMLGG dataset were done. Was a different data set mined?

The GlioVis and cBioPortal analyses were performed using the default settings. Briefly, for *CCL2*, *GDF15* and *ANG* expression and correlation with *ZBTB18* in GlioVis, we mined the TCGA_GBMLGG database; while Kaplan-Meier curves were calculated on the TCGA_GBM

database. We calculated Pearson's correlation, including all histologies and all subtypes. It must be mentioned that in GlioVis *ZBTB18* is still called with his old name (*ZNF238*) in the TCGA database. For cBioPortal analyses, GBM (TCGA, PanCancer Atlas) database was mined for mRNA expression in all samples. We have added all relevant information to the Methods section.

Minor point:

Please provide a rationale for why some experiments are done with some cell lines and others are done with different cell lines. This would help strengthen the logical flow of the paper.

In the initial gene expression analysis, an array of different GBM cells was examined with the goal to robustly identify deregulated cytokine coding genes consistently deregulated upon *ZBTB18*. The validation experiments were then conducted mostly in SNB19 and BTSC233, which do not express *ZBTB18*. BTSC475 was used to knockdown *ZBTB18*, since the gene is expressed at basal levels in this cell line.

Reviewer #2 (Remarks to the Author):

The manuscript analyzed expression of ZBTB18 in different GBM samples and in overexpression/knockdown GBM cell lines. They showed that ZBTB18 affect the expression of a panel of cytokines though potential epigenetic regulation. ZBTB18 promotes glioblastoma progression has been reported by the same group a few years ago. In this manuscript, the authors mainly focused on the regulation of CCL2 and other cytokines by ZBTB18, and showed overexpression of ZBTB18 can increase the infiltration of microglia in mouse tumor xenograft.

1. In Fig1a, the backgrounds of the 5 GBM cell lines are not clearly stated. Please specify which cell line is with or without ZBTB18 overexpression. I also would like to see the western blotting data to show the base line and overexpression of ZBTB18 in all the cell lines used in this figure.

As mentioned above (Reviewer #1), the western blot data for BTSC233, JX6 and SNB19 have been shown in our previous studies (Fedele et al., 2017 Fig. 2C and 2E; Ferrarese et al., 2023, Fig.2A; Masilamani et al., 2022, Fig. 1A, Fig.5A). All these cells do not express ZBTB18. The basal expression of ZBTB18 in various GBM cells established in our group has been shown in our previous study (Masilamani et al., 2017), which also includes a summary table with information about ZBTB18 expression. We have included the WB data of the new experiments (Supplementary Figure 1a-b, BTSC268 and BTSC475) in the revised version of the manuscript.

2. The Fig1c did not have CX3CL1 labeling.

In Figure 1c, only the cytokines that show a statistically significant differential secretion upon ZBTB18 expression have been labeled, as stated in the figure legend. Although we observed a difference in CX3CL1 expression upon ZBTB18 expression, this change does not result in a significant reduction of its secretion, suggesting that other mechanisms are involved. In the following experiments, we decided to focus on the cytokines in which both the gene expression and the protein secretion were affected.

3. The most reliable assay to check the binding activity of ZBTB18 on transcript promotor is to use comparison between ZBTB18 protein (ZBTB18 Abs pulled) vs IgG or Input control in Fig 1F.

Unfortunately, we did not clearly understand this comment. Through all the experiments, the obtained ZBTB18 ChIP values were analyzed and plotted as an enrichment to the input. This is a standard method, which has been used by others and in all our previous published studies. We have clearly stated this procedure in the manuscript.

4. The calculation in the table of Fig.1I seems to be wrong.

We thank the Reviewer for noticing the mistake. We have now included the correct calculation sgZBTB18 #5-4/Ctr in the new Figure 2a.

	Fold change
CCL2	1,452819852
IL8	0,029253524
Angiogenin	0,371208262

5. There is no quantitatively analyzing of correlation between ZBTB18 and GDF15/CCL2.

Unfortunately, it is not clear to which experiment the Reviewer is referring. The correlation between ZBTB18 and CCL2 or GDF15 is shown in Supplementary Figure 2 and commented in the Results section.

6. Based on the plot provided on 2E, one cannot conclude the negative association between ZBTB18 and GDF15/CCL2. Meanwhile, it could be helpful if the authors could label the infiltrating area or tumor core on the images represented.

We have improved Figure 2D-E (now Figure 3d-e) as suggested. We have marked the tumor core (hallmark_hypoxia) and added the correlation values to panel E.

Figure 4. Left panels: stRNA visualization of the hallmark_hypoxia tumor area and the spatial localization of ZBTB18, CCL2 and GDF15 expression in two independent GBM sections. Right panel: Dotplot visualization of the spatial co-localization of the hallmark_hypoxia tumor area and ZBTB18, CCL2 and GDF15 expression. The correlation values are displayed.

7. “Taken together, these data suggest that ZBTB18 exerts a repressive action on the expression and consequent release of cytokines that could play a relevant role in shaping the tumor microenvironment and ultimately, influence the patients’ clinical outcome.” The data presented in the manuscript are not compelling enough to conclude the above statement.

We have rephrased the sentence as: “Taken together, these data suggest that ZBTB18 exerts a repressive action on the expression and consequent release of cytokines that might play a relevant role in shaping the tumor microenvironment and ultimately, influence the patients’ clinical outcome.” to highlight that this is a possibility rather than a fact.

8. SupFig7A, please provide images of Iba1 and F4/80 co-staining.

As suggested, we have included the images of IBA1 staining relative to those showing F4/80 in Supplementary Figure 6a.

Figure 5. Mouse brain sections with tumors derived from BTSC233 EV or BTSC233 ZBTB18 FL xenografts and stained with F4/80 (top panels) or IBA1 antibody (bottom panels); scale bar: 50 μ m.

9. In Fig3A and SugFig7C, why does the intensity of Iba1 decreased in the ZBTB18 overexpression samples? Could the authors provide another microglia marker IF image besides Iba1? And in fig3A, what's the difference between the left and right panel? In Fig.3A, left lower panel, we still could detect a similar amount of IBA+ cells as upper panel indicating BTSC233 EV group. The obvious difference was less intensity of IBA1 florescence. Please explain this difference.

To avoid misinterpretations, in Figure 3A (now Figure 4a), we are now showing IBA1 staining of a representative image of BTSC233 EV- (left) and BTSC233 ZBTB18 FL- (right) derived tumors. Figure 4b represents the quantification of IBA1 signal in all the tumor samples.

Figure 6. Representative micrographs of mouse brain sections with tumors derived from BTSC233 EV or BTSC233 ZBTB18 FL xenografts and stained with IBA1 antibody; scale bar: 50 μ m. (B) Quantification of the immunostaining shown in (A); n=9 biological replicates, *P < 0.05 by a t test.

IBA1 staining intensity is known to be dependent on the activation status of the monocytes (Ito D, Imai Y, Ohsawa K, Nakajima K, Fukuuchi Y, & Kohsaka S (1998). Microglia-specific localisation of a novel calcium binding protein, Iba1. *Molecular Brain Research*, 57(1), 1–9. Sasaki Y, Ohsawa K, Kanazawa H, Kohsaka S, & Imai Y (2001). Iba1 is an actin-cross-linking protein in macrophages/microglia. *Biochemical and Biophysical Research Communications*,

286(2), 292–297.), thus it is possible that the presence of ZBTB18 in GBM cells affected IBA1 expression in TAMs as well. However, the quantification of the labeling was performed by setting a threshold of intensity to which approximately all the red channel (IBA1) signal was detected, in order to take in account different levels of staining across the images.

10. Statistics is needed in Fig.3D.

Unfortunately, we only had a limited number of human samples at our disposal for the analysis described in Figure 3D (now Figure 4d). Thus, the results do not reach statistical significance. For this reason, we decided to show the sample distribution in reference to ZBTB18 and IBA1 staining intensities. In order to assess the reliability of these results, we also performed an *in silico* GlioVis correlation analysis (Figure 3E, now Figure 4e) on the TCGA_GBMLGG database. To better explain the rationale and the results of these experiments, we have rephrased the text in the Results section as follows:

“We then analyzed ZBTB18 and IBA1 expression in a cohort of glioma samples belonging to different CNS WHO grades. The presence of IBA1-positive cells in the samples shows an opposite trend compared to ZBTB18 expression, even though samples where both IBA1 and ZBTB18 appear low could be detected, suggesting that other factors could be implicated in microglia/macrophages recruitment (Figure 4c-d). Given the limited number of biopsies available for the analysis, the statistical power of the experiment was low; thus, in order to implement these results, we performed a GlioVis ²² *in silico* analysis to confirm the negative correlation between ZBTB18 and AIF1 (IBA1) ($r = -0.43$, p Value = 0.00; Figure 4e). ”

11. In Fig4B, it is not convincing that ZBTB18 FL-conditioned microglia grouped separately from the ZBTB18 Nte-SF- and EV-exposed samples, which instead clustered together.

In the principal component analysis in Figure 4B (now Figure 5b), PC1, which explains almost 2 times more variance than PC2, is dividing samples into two groups: EV and the ZBTB18 Nte-SF samples on one side, Ctr and ZBTB18 FL samples on the other side. Therefore, ZBTB18 FL is indeed clustering apart from EV and the ZBTB18 Nte-SF. To support this, we made a consensus clustering partitioning using skmeans (spherical k-means clustering) clustering on top 1, 5, 10, 25, 50 and 75% genes based on ATC with 100 repetitions (as described in <https://academic.oup.com/nar/article/49/3/e15/6020194?login=true>). With $k=2$, one cluster contains samples from EV and the ZBTB18 Nte-SF and the other cluster contains samples from Ctr and ZBTB18 FL. When increasing the number of clusters, we could not identify the 4 conditions. EV and the ZBTB18 Nte-SF remain in the same cluster whereas Ctr and ZBTB18 FL are divided into sub-clusters. We have included this analysis in the manuscript (Supplementary Figure 7a).

Figure 7. Consensus clustering partitioning using skmeans (spherical k-means clustering) clustering on top 1, 5, 10, 25, 50 and 75% genes based on ATC with 100 repetitions.

12. More detailed information is needed for describing for analyzing DEGs in Fig.6B.

We have provided more information regarding Figure 6B (now Figure 7b), both in the results and in the figure. For the top consistent “active” (TCF7L2) and the most inconsistent “inactive” (MEF2C) TF, all known annotated targets from DOROTHEA are shown. Top10 absolute log2 fold changes are labeled. The major active TF identified by Dorothea is TCF7L2, since most of its known upregulated target genes are repressed (i.e. weight > 0 and log2 fold change > 0). Instead, the major inactive TF is MEF2C, since most of its known upregulated target genes are repressed (i.e. weight > 0 and log2 fold change < 0). We have also included a revised panel (below), with a more clear color code.

Figure 8. Plots showing TCF7L2 (left panel) and MEF2C (right panel) associated genes. All known annotated targets from DOROTHEA are shown. Top10 absolute log2 fold changes are labeled. Consistency between the log2 fold change and the prior knowledge (i.e. weight) is color coded.

13. For BTSC475 ZBTB18 KO cell lines, the knockout was confirmed by sanger sequencing, but some normal size ZBTB18 protein are still detected by western. Are these cells a mixed population with wildtype ZBTB18?

In the knockout experiment, single clones were established from a pool of transduced cells; however, it is possible that a small contamination of cells with wildtype ZBTB18 might be present.

14. To compare the infiltration of microglia/macrophage cells in vivo, only ZBTB18 overexpression GBM line is used. I would like to see whether BTSC475 ZBTB18 KO has more infiltrated microglia than BTSC475 ctrl in the xenograft as well.

Unfortunately, we could not perform this experiment, since BTSC475 were not able to induce tumor formation at all. Moreover, as explained below, given the initial limited amount of ZBTB18 to start with, ZBTB18 knockdown experiments are expected to give mild results. In fact, the reason for mostly performing ZBTB18 gain-of-function experiments is that ZBTB18 levels in GBM cells are very low (Fedele et al., 2017; Masilamani et al., 2022). In the literature there are other examples, in which a similar strategy has been adopted to study ZBTB18 in

cancer: ZBTB18 overexpression studies were performed to characterize its role in metastases in breast cancer, in light of the low expression/loss of ZBTB18 in metastatic breast cancer cells (Wang et al., 2023); overexpression studies were also performed to study ZBTB18 function in colorectal cancer, in virtue of its downregulation by DNA methylation (Bazzocco et al., 2021).

15. For the transwell invasion assay, could the authors provide the images of migrated microglia cells besides the statistics analysis? I also would like the authors to add the panel of microglia cells with EV conditioned medium and blocking antibodies of either CCL2 or GDF-15 or both.

We have included the images of migrated microglia (Supplementary Figure 6e). Moreover, as discussed above, we have added results for microglia cells with EV and blocking antibodies for CCL2 and/or GDF15 (Figure 4g).

16. While knocking out ZBTB18 increase CCL2 expression and doesn't affect GDF-15 expression. Is the recruitment of CTBP2 and LSD1 at CCL2 and GDF-15 promoters affected? Do the H3K4me and H3K9me change accordingly?

As mentioned above, due to the low basal level of ZBTB18 in BTSC475, knocking down ZBTB18 has a mild effect. In fact, in the new experiments shown in Figure 2d, ZBTB18 knockout results in a slight upregulation of both CCL2 and GDF15. Moreover, a reduction of ZBTB18 binding at both promoters was observed (Figure 2c).

Figure 9. Enrichment of ZBTB18 ChIP at the *CCL2* (left panel) and *GDF15* (right panel) promoters, in BTSC475sgZBTB18 #5(4). Graphs show the average q-RT PCR results of three independent ChIPs expressed in % input as indicated. Error bars \pm s.d.; * $P < 0.05$ by a t test.

To better define the mechanism of CCL2 and GDF15 regulation we expanded the characterization of ZBTB18 KO phenotype and evaluated whether the histone demethylase LSD1 also plays a role, in virtue of our previous findings showing that ZBTB18 interferes with LSD1-mediated gene activation. Examination of CCL2 and GDF15 expression in BTSC475 showed that the two genes are regulated differently upon LSD1 knockdown. Specifically, we observed downregulation of CCL2 (new Figure 2d), which is in line with our previously reported role of LSD1 as activator of specific ZBTB18 target genes (Ferrarese et al., 2023). In our previous study, we proposed a model in which ZBTB18 repression of its target genes (i.e., SREB genes) occurs through inhibition of LSD1 transcriptional activation function. Conversely, LSD1 silencing induced GDF15 transcription, suggesting that here LSD1 acts as a repressor (new Figure 2d). Furthermore, combining ZBTB18 and LSD1 silencing appeared to have an additive effect (new Figure 2d). The repressive role of LSD1 in the context of GDF15 expression

was further confirmed upon treatment with the LSD1 inhibitor RN-1 (Figure 2e). These new results are shown below in Figure 10.

In conclusion, we propose that LSD1 and ZBTB18 independently participate in the negative regulation of *GDF15* expression.

Figure 10. (A) q-RT PCR analysis of *CCL2* and *GDF15*, upon *ZBTB18* knockdown (sgZBTB18#5(4)) and/or LSD1 silencing in BTSC475. Error bars \pm s.d. of three independent qPCR. Gene expression was normalized to 18s RNA. (B) q-RT PCR analysis of *CCL2* and *GDF15* expression upon treatment with RN-1 in BTSC475. Error bars \pm s.d. of three independent qPCR. Gene expression was normalized to 18s RNA.

We also examined H3K4me and H3K9me change as suggested; however, we decided not to include these data in the revised manuscript. We believe these additional data would make the story overly complicated, given the different role of LSD1 (activator and repressor) in the regulation of the two examined cytokines and the difficulty in clearly attributing the observed changes to LSD1 only or to other uncharacterized factors. Although of interest, a focused characterization of the epigenetic changes at the *CCL2* and *GDF15* promoter should be addressed in a separate study.

17. Does knockout ZBTB18 in GBM cells affect the mice survival or tumor growth after intracranial injection?

As mentioned above, we do not have a proper model to compare tumor formation upon ZBTB18 knockdown, since BTSC475 do not develop tumors in mice. Therefore, we would not be able to interpret the outcome of such an experiment, which, even in the event of tumor formation in some mice of the ZBTB18 knockout group (that could *per se* take several months to grow), would lack a control group for the comparison. As explained above, a knockdown experiment is expected to give only mild results, considering the initial low level of ZBTB18 in the cells. This is in line with tumor suppressors being mostly studied through gain-of-function studies.

18. Fig1F Figure legend should include the full-length ZBTB18 overexpression.

We have now included it.

19. Line 957, redundant “the”

We have corrected the mistake.

20. I would like to see the authors to discuss the contradicting results of ANG in discussion.

We have attempted to provide an explanation for the contradicting results of ANG (see Reviewer #1), which we have included in the discussion.

21. In the Fig1I, the expression changes of CCL2 is 0.69 while IL8 is 34.18. This is a reversed trend to the main text. I'm not sure this is the correct calculation by dividing BTSC475-sgZBTB18 #5 by BTSC475-ctrl.

The Reviewer is correct, we have now changed the panel with the correct calculation (see also comment above from Reviewer #1)

Reviewer #3 (Remarks to the Author):

This manuscript highlights the role of ZBTB18, derived from GBM tumor cells, in regulating expression of TAM chemoattractant CCL2 and GDF15, and in inducing pro-inflammatory phenotypes of microglia, supporting that ZBTB18 overexpression in GBM cells may have therapeutic efficacy in treating GBM. While the regulatory role of GBM-expressing ZBTB18 on TAM recruitment and phenotypical switch is interesting, this manuscript lacks solid experimental evidence to prove the mechanistic hypothesis (mostly based on in silico bioinformatics), which raised serious major concerns that must be addressed as follows:

1. ZBTB18 Nte-SF, a derived product of ZBTB18, is used as another control group in most of the experiments. The results suggested that this byproduct Nte-SF, played an opposite role from ZBTB18. The authors may want to explain (1) why they used this Nte-SF-OV as a independent group, (2) how this byproduct is produced from ZBTB18, (3) the likelihood of introducing this tumor-promotive byproduct into GBM cells while applying ZBTB18 overexpression therapeutic approach, (4) to what extent this byproduct may disrupt the tumor-repressive effect of ZBTB18.

Although already published elsewhere, we agree with the Reviewer that the information about the ZBTB18 SF-Nte and the rationale for including it in some experiments would require additional explanation, which are provided below:

(1 and 2) In our recent study, we have described a new ZBTB18 regulatory mechanism, which involves the proteolytic cleavage of ZBTB18 by the intracellular protease Calpain 2 (Masilamani et al., 2022). The role of the N-terminal fragment was investigated and we demonstrated that the tumor suppressor properties of ZBTB18 full length were lost in the N-terminal cleaved product. Specifically, ZBTB18 SF-Nte no longer localizes into the nucleus and is not able to repress the transcription of previously described ZBTB18 targets (Masilamani et al., 2022; Fedele et al., 2017). On the other hand, ZBTB18 SF-Nte promotes lipid uptake via HIF1alpha and induces a more energetic cell state, suggesting that it might contribute to tumor progression.

In the present study, **ZBTB18 SF-Nte has been included in a subset of experiments to confirm that the observed changes are the consequence of ZBTB18 FL repressive activity and are not mediated by the cleaved fragments.** In fact, we constantly observe that a portion of the ectopic ZBTB18 FL is cleaved, so that the two conditions (FL and SF-Nte) co-exist in the overexpression experiment, possibly masking a ZBTB18-FL specific effect.

(3 and 4) In terms of therapeutic approach, we have shown that the co-existence of the cleaved product does not impair the repressive and tumor suppressive function of ZBTB18 FL, which is mostly based on its ability to repress the expression of various gene categories, which are important for GBM progression, such as mesenchymal genes (Fedele et al., 2017), fatty acid synthesis genes (Ferrarese et al., 2023) and now key cytokines. In fact, the two forms of ZBTB18 are located in distinct cell compartments (FL in the nucleus, SF-Nte in the cytoplasm). However, ZBTB18 SF-Nte also induces new functions (i.e. lipid uptake), which might support the tumor growth. Therefore, as a potential therapeutic application, it would be important to counteract/limit ZBTB18 cleavage in order to efficiently and safely deliver it to the tumor cells.

2. The authors constantly used gain-of-function (ZBTB18 overexpression) experiments to demonstrate the downstream functions (e.g., the negative regulatory effect of ZBTB18 on chemoattractant CCL2 and GDF15), while loss-of-function experiments were rather lacking throughout the manuscript. ZBTB18 KO even exerts no change on GDF-15. These raised serious concern about the potency of regulatory effect from ZBTB18 on the proposed downstream functions.

The reason for mostly performing ZBTB18 gain-of-function experiments is that ZBTB18 levels in GBM cells are very low (Fedele et al., 2017; Masilamani et al., 2022). Similarly, overexpression studies were performed to characterize the role of ZBTB18 in metastases in breast cancer, in light of the low expression/loss of ZBTB18 in metastatic breast cancer cells (Wang et al., 2023). Overexpression studies were also performed to study ZBTB18 function in colorectal cancer, in virtue of its downregulation by DNA methylation (Bazzocco et al., 2021). In our studies, BTSC475 were chosen for conducting complementary loss-of-function experiments since those cells express slightly more ZBTB18 compared to other cells (Masilamani et al., 2022; Ferrarese et al., 2023). However, given the initial limited amount of ZBTB18 to start with, ZBTB18 knockdown experiments are expected to give mild results and were mostly used to confirm the findings of the gain-of-function experiments. Please note that the western blot in Fig. S5c nuclear extracts were examined to allow a better detection of ZBTB18.

Nevertheless, we have expanded the set of loss-of-function experiments, which are included in the new Figure 2. In particular, we have confirmed ZBTB18 role as a direct repressor of both CCL2 and GDF15, as demonstrated by qRT-PCR (Figure 2d) and ChIP (Figure 2c). See also Figure 9 above and reply to Rev#2). Moreover, we have investigated the role of LSD1 silencing, alone or in combination with ZBTB18, in virtue of our previously described ZBTB18 role in halting LSD1-mediated gene activation (Ferrarese et al., 2023). As explained above (Rev. #2, Figure 10), examination of CCL2 and GDF15 expression in BTSC475 showed that the two genes are regulated differently upon LSD1 knockdown. Specifically, we observed downregulation of CCL2 (new Figure 2d), which is in line with our previously reported role of LSD1 as activator of specific ZBTB18 target genes (Ferrarese et al., 2023). In our previous study, we proposed a model in which ZBTB18 repression of its target genes (i.e., SREB genes) occurs through inhibition of LSD1 transcriptional activation function. Consistent with that model our new data show that CCL2 re-expression induced by ZBTB18 knockdown requires LSD1 (new Figure 2d). Conversely, LSD1 silencing induced GDF15 transcription, suggesting that here LSD1 acts as a repressor (new Figure 2d). Furthermore, combining ZBTB18 and LSD1 silencing appeared to have an additive effect (new Figure 2d). The repressive role of LSD1 in the context of GDF15 expression was further confirmed upon treatment with the LSD1 inhibitor RN-1 (Figure 2e) (Rev. #2, Figure 10B). We have also performed new ELISA experiments in the same

experimental setting, which revealed a different effect of ZBTB18 loss on CCL2 and GDF15 secretion. Our new data, included below (Figure 11-12) and in the new version of the manuscript (Figure 2b, f-h), revealed a more complex role of ZBTB18 in GDF15 processing and secretion. We provided a detailed discussion in the manuscript and acknowledge that would be worth following up in separate study.

Figure 11. CCL2 (left panel) and GDF-15 (right panel) ELISA showing the amount of the secreted proteins in the conditioned medium of BTSC475 transduced with sgZBTB18 #5(4) and/or shLSD1. n=4 biological replicates; error bars \pm s.d.; *P < 0.05, **P < 0.01, and ***P < 0.001 by a t test.

Figure 12. (Left panel) Western blot analysis of pro- and mature GDF-15 in BTSC475 transduced with sgZBTB18 #5(4) and/or shLSD1. α Tubulin is used as a loading control. (Right panel) Quantification of pro- and mature GDF-15 (normalized to α Tubulin).

Finally, we would like to highlight that ZBTB18 overexpression and knockout led to a clear opposite phenotype when examining the lipid droplet content of microglia (Figure 7d-e), further supporting our conclusion regarding the role of ZBTB18 in GAM phenotype.

3. The authors indicate the topic in the title as glioblastoma; however, when proving clinical relevance of ZBTB18, the analyzed datasets included LGG as well (in Figure S2B, Figure 3E, etc.). Will the significance be impaired if datasets only containing GBM samples were analyzed?

The potential clinical relevance of ZBTB18 in GBM has been extensively evaluated in previous publications (Carro et al., 2010; Fedele et al., 2017; Masilamani et al., 2022; Ferrarese et al., 2023) and it was not the focus of this article. In the specific context discussed here, different tumor grades were included to examine the correlation between ZBTB18 and IBA1 in a set of samples showing variable ZBTB18 expression (i.e., high in low grade tumors and low in high

grade tumors). Noteworthy, a negative correlation between ZBTB18 and CCL2 is observed also when examining datasets only containing GBM samples.

Figure 13. Pearson correlation analysis between *ZBTB18* and *CCL2* expression in the TCGA_GBM dataset.

4. The mechanistic study of ZBTB18 transcriptionally regulating CCL2 and GDF15 is not sufficient. (1) Line 306: EMSA is suggested to prove direct binding of ZBTB18 at the CCL2 and GDF15 promoter. (2) What promotes ZBTB18 to be capable of directly regulating several chemokines? (3) The authors may also design rescue experiments to demonstrate that ZBTB18 transcriptionally represses CCL2 and GDF15 via dysregulating histone demethylase activity.

ZBTB18 is a transcriptional repressor, which binds to promoter regions enriched for ZNF238 (ZBTB18) binding motif, as previously demonstrated by others and us (Fedele et al., 2017; Ferrarese et al., 2023, Wang et al., 2023; Xie et al., 2021). In those studies, CHIP has been successfully used to identify ZBTB18 direct targets, which have been shown to contain ZBTB18 binding motifs.

In order to answer the Reviewer’s comment, we searched for ZNF238 motifs in motifDB. We found a few quite consistent motifs in several databases. Counting for the occurrence of each motive with at least 90% homology in a region +/- 500bp around TSS of selected genes led to the identification of the hPDI and HOCOMOCO motifs in pretty much all transcripts (see table below). However, it is important to keep in mind that, although this analysis supports the notion that ZBTB18 regulates gene expression by direct binding to the promoter region, the presence of a motif does not necessarily means that the TF can bind and the.

	Hsapiens.HOCOMOCOv11.core. C.ZBT18_HUMAN.H11MO.0.C	Hsapiens.hPDI.ZNF238
CCL2_196567	2	3
CCL2_196568	2	3
GDF15_215850	4	0

GDF15_215851	4	0
GDF15_215852	6	3
GDF15_215853	3	5
GDF15_215855	3	5

In addition to a direct gene repression, we have previously shown that ZBTB18 can also interact with other factors (i.e., CTBP2 and LSD1), limiting their transcriptional regulation capability (Ferrarese et al., 2023). As explained above (see answer to comment #2 and comment #11 from Rev. #2), we have now performed additional experiments to further elucidate the mechanism of ZBTB18-mediated regulation of *CCL2* and *GDF15*. According to our model, LSD1 acts as a transcriptional activator of *CCL2* and this activity is counteracted by ZBTB18 in line with our previously described model (Ferrarese et al., 2023). Instead, at the *GDF15* promoter, LSD1 knockout results in *GDF15* upregulation and this effect is further enhanced upon ZBTB18 knockdown. Interestingly, treatment with the LSD1 inhibitor RN-1 also leads to *GDF15* upregulation. Therefore, we propose that LSD1 and ZBTB18 independently participate in the regulation of *GDF15*. Concerning the regulation of histone demethylase activity, we have performed the indicated experiments but we decided not to include these data in the revised manuscript. We believe these additional data would make the story overly complicated, given the different role of LSD1 (activator and repressor) in the regulation of the two examined cytokines and the difficulty in clearly attributing the observed changes to LSD1 only or to other uncharacterized factors. Although of interest, a focused characterization of the epigenetic changes at the *CCL2* and *GDF15* promoter should be addressed in a separate study.

Other concerns:

1. The correlation between ZBTB18 and its downstream effectors, as revealed by bioinformatics in Figure S2B, Figure 3E, should be validated experimentally.

We respectfully disagree; the purpose of this analysis is to validate in a larger patient cohort the findings of our in-house experiment shown in Figure 4c-d (previously Figure 3C-D). Therefore, the *in silico* analysis is already an implementation to support the observed inverse relationship between ZBTB18 and AIF1/IBA1.

2. Line 333: how did the authors draw the conclusion that ZBTB18 expression is localized to GBM cells as no other non-GBM cells was displayed in Figure 2A.

In the text, we do not mean to infer that ZBTB18 is only localized to GBM cells, but rather we point at which subtype of GBM cells ZBTB18 localizes with. The identification of ZBTB18 expression within the infiltrative lineage-like cells (Oligodendrocyte progenitor cell like (OPC-like) and Neuronal progenitor cell like (NPC-like cells) but its absence in the Mesenchymal like (MES-like) cells is consistent with our previous identification of ZBTB18 as a repressor of mesenchymal genes in GBM (Carro et al., 2010; Fedele et al., 2017).

3. Line 385: no show of ZBTB18 Nte-SF group in Figure 3F.

We thank the Reviewer for noticing the mistake; we have removed ZBTB18 Nte-SF.

4. Line 408/Figure 4B: possible overstatement. No clear discrepancy among the red, blue and green dots (ctrl, EV, ZBTB18 FL).

As mentioned above (see answer to comment 11 from Rev#2), the principal component analysis in Figure 4B (now Figure 5b), shows that indeed, ZBTB18 FL is clustering apart from EV and the ZBTB18 Nte-SF. To support this, we made a consensus clustering partitioning using skmeans (spherical k-means clustering) (new Supplementary Figure 7a), which is in line with the previous observation.

5. Figure 5H: not convincing at all. What is the situation under in vivo conditions?

Unfortunately, it is not clear what the Reviewer is criticizing. We have performed MHCII staining on conditioned microglia because it was our goal to validate GMap data that suggested an enrichment of the TAM-BDM MHC module in microglia conditioned with the medium of ZBTB18 FL expressing cells. Therefore, we tested the protein expression by immunofluorescence in the same experimental conditions. The MHCII staining in our biological replicates is accompanied by a quantification of the results, presented in the results in new Figure 6i. This indicates an overall increase in MHCII staining in microglia upon exposure to the conditioned medium of ZBTB18-expressing BTSC233. This result is consistent with the upregulation of MHCII (HLA-DMA) detected by RNAseq:

symbol	entrez	logFC	AveExpr	P.Value	adj.P.Val
HLA-DMA	3108	3,4666608	3,33139434	0,00122797	0,0203351

Regarding the Reviewer's suggestion about investigating the expression of MHCII *in vivo*, although certainly interesting, it might require the setup of an additional array of experiments that goes beyond the scope of the present research. In fact, in our *in vivo* model we observed a reduction of recruited GAMs to the tumor site; however, it is not clear whether the few GAMs that are still present after ZBTB18 overexpression are escapees that managed to infiltrate the tumor anyway, or they indeed show a more pro-inflammatory commitment, as the one highlighted in our RNAseq experiment. Addressing this relevant question about not only MHCII expression, but also, more in general, of the GAM commitment upon ZBTB18 overexpression *in vivo*, is crucial and it will definitely be tackled in the follow-up of this research.

Reviewers' comments:

Reviewer #1 (Remarks to the Author):

The authors have satisfactorily addressed many of the reviewers' comments and added new data and clarifications that strengthen the manuscript. However, some concerns remain.

Specific points:

1) As mentioned in the first round of review, "it would be important to provide the tumor mass/size for each mouse and the expression levels of ZBTB18 in the tumors. With the information provided currently, it is difficult to assess whether the changes in microglia/macrophages are directly correlating with changes ZBTB18 levels or whether differences in tumor size/stage also contribute to the phenotype".

These points have not been convincingly addressed by the authors. They indicated that tumors have comparable sizes and referred to Fedele et al., 2017 Fig.4C-D for data showing this. However, there is no measurement of tumor size in those figures. Could the authors provide a quantitation? It would also be relevant to confirm the expression levels of ZBTB18 in those tumors.

We have included pictures showing the immunostaining of ZBTB18 to confirm its expression in the tumors originated from the cells overexpressing it (panel a-b). We have also included the quantification of the tumor area based on H&E (panel c). These data have been included in the new Supplementary Figure 9.

Figure 1. (a) Representative micrographs of mouse brain sections with tumors derived from BTSC233 EV or BTSC233 ZBTB18 FL xenografts and stained with ZBTB18 rabbit polyclonal (#12714-1-AP, Proteintech); scale bar: 50 μ m. (b) Quantification of the immunostaining shown in (a); n=4 biological replicates; *P < 0.05 by a t test. Quantification of tumor derived from BTSC233 EV or BTSC233 ZBTB18 FL xenografts based on H&E staining (representative stainings were shown in Fedele et al., 2017). n=9 (EV) and n=7 (ZBTB18-FL) biological replicates; ns (not significant).

2) Line 339-342: The discussion of supplementary figure 3 should be modified. Although fitting curves on this data suggests a negative correlation, the biological relevance of this conclusion is questionable. Indeed, as shown previously by the authors, the vast majority of GBM express very low levels of ZBTB18 (if any). Thus, any correlation between ZBTB18 expression levels and that of other markers is very difficult to make. The conclusion here should be that various levels of CCL2 are expressed by GBM, which have low levels of ZBTB18 expression.

We agree with the Reviewer and we have rephrased the conclusion accordingly. We have included a final remark that incorporates the analyses from GBMLGG and GBM only (requested by Reviewer#3) (new Supplementary Figure 3):

“The negative correlation between ZBTB18 and these cytokines was detected even within a dataset including GBM only; however, since ZBTB18 is almost uniformly absent from GBM, the statistical power of these analyses is consequently weaker. Within this dataset, we observed a uniformly low expression of ZBTB18, compared to different levels of cytokine expression (Supplementary Figure 3b-c)”.

3) Figure 2. The effect of ZBTB18 knockout on CCL2 protein expression level is not convincing. Specifically, while a slight increase is seen in Fig 2a and 2b (with clone 24), no difference is seen in Fig 2b (with clone 4) and 2f, despite these experiments all being a very similar. This should be better explained in the summary of this section on lines 400-403.

Furthermore, the confirmation of knockout for clone 24 should be shown.

We agree with the Reviewer and our conclusion is that, since the level of secreted CCL2 is very high in the control BTSC475 (as can be observed in the cytokine array in panel a), the knockdown of ZBTB18 produces only a mild effect. The results shown in Fig.2b using two ZBTB18 ko clones give a similar result, however the p value for ZBTB18ko#5(4) is 0,0615 and could not be considered significant. We have more clearly described the outcome of this experiment. For the same reason, we did not observe a clear increase of CCL2 secretion by ELISA in the experiment shown in Fig. 2f. Here, the goal is to show the effect of LSD1 knockout; however, since it was important to test the effect of ZBTB18 and LSD1 loss in the context of GDF15 secretion, we opted for showing a consistent experiment (ZBTB18 or LSD1 ko, alone or in combination) for both cytokines, CCL2 and GDF15. We have added the comments to explain why an increase of CCL2 secretion is not observed upon ZBTB18 ko:

Lines 395-397: “We speculated that, since the level of secreted CCL2 is very high in the control BTSC475 (as can be observed in the cytokine array in Figure 2a), the knockdown of ZBTB18 produces only a mild effect”.

Lines 430-432: “(Figure 2f, left panel), even though the concentration of CCL2 in BTSC475 Ctrl and sgZBTB18 #5(4) was very high and thus, resulting in a possible underestimation of the actual cytokine amount”.

Regarding the confirmation of ZBTB18#4(24) (sequence #4, clone 24), this clone was established from the ZBTB18 ko#4, which was described in our previous publication (Ferrarese et al., 2023 FigS6). We are including the figure below (Figure 2) for the Reviewer’s convenience. A western blot showing ZBTB18 expression in the clone#24 is shown below (Figure 3) and has also been included in Supplementary Figure S6.

Figure 2. Supplementary Figure S6 (panels a-b) from Ferrarese et al., 2023, which shows the characterization of BTSC233 with sgZBTB18#4.

Figure 3. Western blot showing the expression of sgZBTB18#4 clone#24 used for the experiment in Figure 2b.

4) Fig. 3d. A scale bar for the meaning of the different colors could be helpful.

We agree with the suggestion and we have included a scale bar to describe the colors used in Figure 3d.

5) Fig 3e. The data in Fig.3e show a lack of correlation between ZBTB18 expression and the other markers (values are close to 0). Inverse correlations would show value close to -1. Therefore, the authors should change the wording of their conclusion (lines 410-413) to indicate a lack of correlation.

We agree with the Reviewer and we have rephrased the sentence as follows (lines 461-465):

“A **lack of association** with ZBTB18 expression was also confirmed by the spatially resolved RNA sequencing data (Figure 3d, Supplementary Figure 8d), suggesting that the alteration of IL8 and angiogenin secretion upon ZBTB18 perturbation is not directly linked to ZBTB18 transcriptional repressive role.”

6) Supplementary figure 2b: Please add statistics.

Statistics were included in the Results paragraph. We have included a table summarizing the statistics of the Pearson correlation shown in Supplementary 3a-b (TCGA_GBMLGG and TCGA_GBM) in Supplementary Figure 3 (panel c).

Reviewer #2 (Remarks to the Author):

The authors have addressed all my concerns.

Reviewer #3 (Remarks to the Author):

These concerns remain to be addressed:

1. For major concern 1: the authors admitted that the two forms of ZBTB18 (with opposite functions) co-exist, but they did not indicate where are the supporting data for “In terms of therapeutic approach, we have shown that the co-existence of the cleaved product does not impair the repressive and tumor suppressive function of ZBTB18 FL,...”

We apologize to the Reviewer if his previous concern ((3) the likelihood of introducing this tumor-promoting byproduct into GBM cells while applying ZBTB18 overexpression therapeutic approach) was not adequately addressed. We believe that, since the tumor promoting effects of ZBTB18 are related to its nuclear localization and consequent repression of genes involved in GBM functions (mesenchymal genes (Fedele et al., 2017), fatty acid synthesis genes (Ferrarese et al., 2023), and now tumor promoting cytokines), the presence of the SF-Nte does not interfere with the activity of the ZBTB18 long form. In fact, ZBTB18 long form seems to have a dominant effect when both forms are present. In particular, the ZBTB18 SF-mediated effects were observed only when this form was ectopically expressed, in absence of ZBTB18 FL (Masilamani et al., 2022). This hypothesis is confirmed by the previously reported tumor suppressive effect of ZBTB18 observed in various phenotypic assays (cell proliferation, cell migration and invasion), as well as *in vivo* (suppressed or delayed tumor formation) (Fedele et al., 2017). However, in light of possible future therapeutic approaches aimed at re-expressing ZBTB18 in GBM cells, it would be important to completely suppress the expression of the cleaved product, to avoid any unexpected, adverse effect. This could be achieved by blocking Calpain 2 or by engineering the cleavage site; we will try to follow this approach in future studies. We have included a final comment in the revised manuscript (lines 789-792).

2. For major concern 3: the authors did not answer my questions. What would the correlation be like in Fig. S2b and the revised Fig. 4e if the TCGA_GBM dataset rather than TCGA_GBMLGG was applied for analysis? Will it still be significant?

We have included the plots requested by the Reviewer (please see the picture below) as new Supplementary Figure 3b-c. However, as correctly pointed out by Reviewer #2, it is important to consider that, since ZBTB18 is almost uniformly absent from GBM, the statistical power of these analyses is weaker. We have included this comment in the revised manuscript (lines 363-368).

b

	TCGA_GBM		TCGA_GBMLGG	
ZBTB18 correlation vs.	r	p	r	p
CCL2	-0,26	e-6,26	-0,5	0
GDF15	-0,37	e-9,27	-0,62	0
ANG	-0,39	e-9,68	-0,61	0
AIF1	-0,16	e-3,66	-0,43	0

Figure 4. Pearson correlation analysis between *ZBTB18* and *CCL2*, *GDF15*, *AIF1*, or *ANG* expression in the TCGA_GBM dataset. The corresponding statistics are shown in the lower table.

3. For major concern 4: (1) missing words in the sentence of the second paragraph? (“the presence of a motif does not necessarily means that the TF can bind and the.”) (2) The authors did not perform the

suggested experiments or did not show the results in the point-by-point response letter which they decided not to include in the revised manuscript.

(1) We have included the missing word (promoter).

(2) Regarding the previous points of major concern #4, we apologize for not clearly addressing the points, as we mostly focused on the description of the further characterized mechanism. Since we are not sure whether all or only part of the comments were not fully addressed, we are replying to all the Reviewer's concern here:

(1) Line 306: EMSA is suggested to prove direct binding of ZBTB18 at the CCL2 and GDF15 promoter.

We respectfully disagree with the Reviewer regarding the requirement of an EMSA assay. The ChIP experiments performed with ZBTB18 overexpression and upon ZBTB18 knockdown show that ZBTB18 can bind to the promoter of *CCL2* and *GDF15*, in proximity to the TSS. ChIP is the gold standard technique used to demonstrate the direct binding of a transcription factor to a gene region (i.e. promoter) in an intact cell system, without introducing external components. In fact, in an EMSA transcription factors and DNA are combined *in vitro*, which could produce artifacts. EMSA has clearly the advantage of being a sensitive technique, which also allows testing different DNA sequence length and conformation, however, this goes beyond the depth of mechanistic insight aimed with this study.

(2) What promotes ZBTB18 to be capable of directly regulating several chemokines?

As we wrote earlier, ZBTB18 is a transcriptional repressor, which binds to promoter regions enriched for ZNF238 (ZBTB18) binding motif, as previously demonstrated by others and us (Fedele et al., 2017; Ferrarese et al., 2023, Wang et al., 2023; Xie et al., 2021). In those studies, ChIP has been successfully used to identify ZBTB18 direct targets, which have been shown to contain ZBTB18 binding motifs. In order to answer the Reviewer's comment, we searched for ZNF238 motifs in motifDB. We found a few quite consistent motifs in several databases. Counting for the occurrence of each motive with at least 90% homology in a region +/- 500bp around TSS of selected genes led to the identification of the hPDI and HOCOMOCO motifs in pretty much all transcripts. However, it is important to keep in mind that, although this analysis supports the notion that ZBTB18 regulates gene expression by direct binding to the promoter region, the presence of a motif does not necessarily mean that the TF can bind to the promoter.

To better address this point, we have now included a scheme of the promoter and a table with the predicted binding site in the new Supplementary Figure (5) and better discussed the previously included ChIP results (lines 378-382).

(3) The authors may also design rescue experiments to demonstrate that ZBTB18 transcriptionally represses CCL2 and GDF15 via dysregulating histone demethylase activity.

We think that the Reviewer is also asking about the experiments aimed at elucidating whether ZBTB18 affects LSD1 demethylase activity on *CCL2* and *GDF15* promoter regulation, which we decided not to show, since we believe these additional data are not sufficient to characterize the mechanism of ZBTB18 and LSD1 function, furthermore, expanding the study in this direction would make the story overly complicated. In fact, given the different role of LSD1 (activator and repressor) in the regulation of the two examined cytokines, it is difficult to clearly attribute the observed changes to LSD1 only or to other uncharacterized factors.

As requested, we are showing the data here:

We also investigated whether ZBTB18 loss affects the presence of H3K4me2 and H3K9me2 at the two promoters. As shown below, we observed a decrease of H3K4me2 marks at both *CCL2* and *GDF15* promoters, while no changes about H3K9me2 abundance (which is overall quite low) were observed

(Figure 5). While the H3K4me2 decrease at the *CCL2* promoter could be explained by the restored LSD1 inhibition in absence of ZBTB18, for *GDF15* we do not have a convincing explanation.

Figure 5. Enrichment of H3K4me2 and H3K9me2 ChIP at the *CCL2* (left panels) and *GDF15* (right panels) promoters, in BTSC475 sgZBTB18#5(4). Graphs show the average q-RT PCR results of three independent ChIPs expressed in % input as indicated. Error bars \pm s.d.; *P < 0.05 by a t test.

Overall, we believe that the data discussed here suggest a complex gene expression regulatory mechanism and that more experiments (i.e. ChIPseq) need to be performed to clearly understand the modality of *CCL2* and *GDF15* regulation by ZBTB18 and LSD1. For this reason, we have not included these data in the revised manuscript. We have clearly indicated the limit of this study at the end of ZBTB18-LSD1 regulatory mechanism discussion: *“In this perspective, also a deeper investigation of the complex ZBTB18 and LSD1-mediated regulatory mechanism of GDF15 and CCL2 expression, including changes of histone marks at the respective promoters, seems worth addressing in the future.”* (Lines 698-701).

Nevertheless, to give additional mechanistic insight about the role of LSD1, we performed new ChIP experiments in GBM#22 cells with LSD1 ko (described in Ferrarese et al., 2023 and Faletti et al., 2022), and tested whether LSD1 affects the presence of H3K4me2 and H3K9me2 marks, consistent with its histone demethylase activity. The results shown below (Figure 6) and included in the new Supplementary Figure 7, clearly indicate an enrichment of H3K4me2 at the *GDF15* promoter only, which is consistent with the repressive role of LSD1 in *GDF15* regulation shown in Figure 2. Instead, no change was observed at the *CCL2* promoter suggesting that LSD1 activates *CCL2* expression but this does not require its histone demethylase activity, as also indicated by the lack of effect upon RN1 treatment (Figure 2e and Supplementary Figure 7b). It is important to notice that the primers used for the qPCR reaction are the same used in the ZBTB18 or HA ChIP and map close to the TSS. Whether ZBTB18 and LSD1 belong to the same complex, possibly including other factors, will be the object of future studies.

Figure 6. Enrichment of H3K4me2 and H3K9me2 ChIP at the *CCL2* (upper panels) and *GDF15* (lower panels) promoters, in BTSC165 sgLSD1. Graphs show the average q-RT PCR results of three independent ChIPs expressed in % input as indicated. Error bars \pm s.d.; *P < 0.05 by a t test.

4. For other concern 5: low quality and unconvincing images presented in Figure 5H. No more convincing data (e.g. flow cytometry, IHC staining of slides from *in vivo* models) were included in the revision.

In order to test the expression of MHCII upon ZBTB18 expression *in vivo*, we first attempted to stain brain sections of the BTSC233 xenograft model (Figure 4 and Fedele et al., 2017). However, no clear MHCII signal was observed, most likely due to the lack of adaptive response in the NOD/SCID mouse model adopted. We then opted for measuring MHCII expression in a recently established immunocompetent xenograft model established in our laboratory. Specifically, murine GBM cells (KAB203) derived from a previously described GBM mouse model (RCAS PDGFA/shpTrp53) (Gift from Dr. Squatrito; Squatrito et al, personal communication; Hambardzumyan et al., 2009), were transduced with a control vector or with a ZBTB18 expressing lentivirus and injected in C57BL/6 mice.

Tumors were detected in 7 out of 9 mice in the EV group and 6 out of 9 mice in the ZBTB18 group. We then compared IBA1 and MHCII expression: a slight increase of IBA1 expressing cells was observed, although not statistically significant, possibly due to the limited number of samples together with the high variability of IBA1-positive cells in ZBTB18 expressing tumors (new Supplementary Figure 12g-h). Immunostaining of MHCII indicated a higher expression in the ZBTB18 group (new Figure 7d-e), further supporting our previous observation. The new data are included in the figure 7 and 8 below:

Figure 7. (a) Western blot showing ZBTB18 FL expression in KAB203 cells transduced with control (EV) or ZBTB18 FL-expressing lentivirus. (b) Microphotographs of H&E stained brain sections derived from C57BL/6 mice injected with KAB203 cells (transduced with EV or ZBTB18 FL). Images were selected from tumors, which showed comparable characteristics in terms of size and location; scale bar: 1 mm. (c) Quantification of the area of the brain sections shown in (b). (d) Representative micrographs of mouse brain sections with tumors derived from

KAB203 (transduced with EV or ZBTB18 FL) syngeneic mice and stained with ZBTB18 antibody; scale bar: 50 μ m. (e) Kaplan-Meier survival curve of mice injected with KAB203-EV or KAB203-ZBTB18 mice (n=9). No difference in survival was observed ($p=0.951$). (f) Representative H&E stained brain section derived from KAB203-ZBTB18 injected mice, showing the presence of tumor cells along the meninges; scale bar: 1 mm. (g) Representative micrographs of mouse brain sections with tumors derived from KAB203 (transduced with EV or ZBTB18 FL) syngeneic mice and stained with IBA1 antibody; scale bar: 50 μ m. (h) Quantification of the immunostaining shown in (f); n=6 (EV), n=3 (ZBTB18 FL) biological replicates; * $P < 0.05$ by a t test. Dark dots correspond to the samples shown in (b) and (c).

Figure 8. (Top panels) Representative micrographs of mouse brain sections with tumors derived from KAB203 EV or KAB203 ZBTB18 FL cells and stained with MHCII antibody; scale bar: 50 μ m (top panels) and 25 μ m (insets, bottom panels). (Bottom panel) Quantification of the immunostaining shown in (d); n=6 (EV) and n=3 (ZBTB18 FL) biological replicates; * $P < 0.05$ by a t test.

Moreover, extracted murine microglia with the CD11b beads system (Miltenyl Biotech) followed by RNA isolation. We were able to obtain microglia RNA from two tumor samples, one EV (#47) and one expressing ZBTB18 FL (#44), which had a similar IBA1 content, respectively. qPCR analysis showed an

upregulation of both HLA alleles (H2 Aa and H2 Ab), which is in line with our staining and the RNAseq data obtained from conditioned microglia.

Figure 9. q-RT PCR analysis of *H2Aa* and *H2Ab* expression, in the indicated KAB203-derived tumors. Gene expression was normalized to mouse Hprt. n=3 technical replicates; error bars \pm s.d.

We have included these data in the new Figure 7 and new Supplementary Figure 12. The decision to include this set of experiments in a new mouse model was taken to be able to address the Reviewer's request of further characterization of MHCII expression in microglia *in vivo*, in light of the impossibility to show this in our previously used immunodeficient mouse model. However, the KAB203-C57BL/6 model significantly differs from the NOD/SCID one used in the rest of the manuscript, namely in the fact that it is a syngeneic mouse model with a fully functional immune system. This might explain the observed differences in terms of survival and macrophages/microglia infiltration. It is then crucial to expand the research on this model in order to validate these results and to increase the statistical power of the analyses. The full characterization of such a model will be key to understanding how the tumor microenvironment is able to interact with a functional host immune system, which will be the object of our future research.

We have included a comment on the subject in the Discussion section.

Reviewers' comments:

Reviewer #1:

- 1) Sup. Fig. 12a: A loading control is missing. Please include a reprobing of the same membrane with anti-Tubulin.

Reply: The Reviewer is correct; we apologize for the overlook. We have included the tubulin control in Supplementary Figure 12a.

- 2) Sup. Fig. 12b, c: Why were only 3 EV tumors included in the analyses? The authors mentioned that 7 out of 9 EV tumors grew and that “one tumor in the EV group had cells growing both in the striatum and outside the cortex, and therefore was not included in the subsequent examinations”. Thus 6 tumors should remain and be included in subsequent analyses presented in Figure 7d, e and Supplementary Figure 12 b, c, d, g, h.

Furthermore, while the manuscript mentions that only 3 out of 9 tumors grew in the ZBTB18 FL group, the rebuttal letter mentions that 6 out of 9 tumors from this group grew. Please make sure the correct information is reported in the manuscript.

Reply: The Reviewer has correctly understood that the tumors included in our analyses were 6 in the EV group and 3 in the ZBTB18 FL group. However, we realize that the organization of the panels was probably misleading. To clarify further, Supplementary Figure 12b-d shows a pool of 3 EV and 3 ZBTB18 FL tumors with similar aspect and size (summarized by the quantification in Supplementary Figure 12c). Supplementary Figure 12b shows the overall aspect of the tumors, while Supplementary Figure 12d confirms the presence of ZBTB18 in the tumors derived from the cells ectopically expressing it (all 3 ZBTB18 expressing tumors were included). Regarding MHCII staining (Figure 7d-e) and IBA1 staining (Supplementary Figure 12g-h), we included the representative pictures of the tumors shown in Supplementary Figure 12b (MHCII: Figure 7d; IBA1: Supplementary Figure 12g); however, when we quantified these two staining (MHCII: Figure 7e; IBA1: Supplementary Figure 12h), we included all the 6 EV tumors and the 3 ZBTB18 FL ones. To allow an easier identification of the EV tumors shown in the representative pictures, we highlighted their corresponding dots in black, while those not shown in the staining panels were colored gray. The number of biological replicates (6 EV, 3 ZBTB18 FL) was also indicated in the figure legend, and we have now included it also in the manuscript body.

Regarding the discrepancy in the number of ZBTB18 FL tumors between the rebuttal letter and the manuscript, we apologize for the mistake; the correct information is the one in the manuscript (3 out of 9).

- 3) Sup. Fig. 12f: Please better explain where the meninges are and where interaction with the tumor cells can be seen. A higher magnification image and/or an arrow might be helpful.

Reply: We thank the Reviewer for the suggestion. We are including a higher magnification inset of the area of interest and highlighting the relative positions of the tumor cells and the meninges.

- 4) In the BTSC233 model (in NSG mice), ZBTB18 suppresses tumor growth, but these tumors do not express MHCII. In contrast, the new data with the KAB203 model (in immunocompetent mice) shows that ZBTB18 increases MHCII expression, but it remains unclear whether ZBTB18 affects tumor growth and survival in this model. Taken together, data from these two models argue against MHCII playing an important role in the anti-tumorigenic effect of ZBTB18. How can these results be reconciled with the central message of the manuscript, which proposes that ZBTB18 regulates microglia-tumor cell interactions and that this presumably affects tumor growth?

Reply: We thank the Reviewer for this important comment. However, we respectfully disagree with the conclusion. In the new model (KAB203) we indeed observe a difference in the number of tumors that arise in the control (7/9) and ZBTB18-FL (3/9); however, in some cases, tumor cells grew in the subarachnoid space outside the meninges causing a premature death of these animals. Thus, it is not possible to accurately determine the survival rate in the two conditions. As we wrote in the previous rebuttal letter, this experimental model has been included to address Rev#3 concern about a change in MHCII levels upon ZBTB18 expression. Although we were able to confirm the expression of MHCII in ZBTB18 expressing tumors, this model is not yet sufficiently characterized to allow further speculations. In fact, more studies should be performed to determine the role of ZBTB18 in the establishment of the tumor microenvironment in presence of a fully functional immune system. We believe that this goes beyond the scope of this manuscript.

We would also like to clarify that the central conclusion of our manuscript is not that ZBTB18 regulates tumor growth through MHCII but that ZBTB18 affects the phenotype of microglia, which acquires more inflammatory features. In this context, an increase in MHCII expression was initially observed in microglia cells conditioned with BTSC233 cells expressing ZBTB18-FL by RNAseq and later confirmed by immunostaining. However, the situation *in vivo* is likely more complex and, considering the lack of an intact immune system, NOD/SCID mice might not represent the best fitting model to study microglia/macrophage commitment.

We think that each model allows the characterization of different aspects of the microglia changes induced by ZBTB18 expression in GBM cells. In fact, while a strong effect on microglia migration can be observed in the BTSC233 model, the KAB203 model confirms *in vivo* the induction of MHCII, which was previously only observed *in vitro* and which is consistent with a change in the microglia phenotype highlighted by RNAseq. Supporting our results, previous studies have shown that the presence of T cells is important for MHCII expression in microglia (Subbarayan et al., 2002) and this could explain why the same effect was not observed in the NOD/SCID mouse system. In the future we plan to investigate more in depth the interplay of microglia and T cells in the presence of ZBTB18.

We have added a comment to this issue in the Discussion section.

Reviewer #1 (Remarks to the Author):

The authors have satisfactorily addressed most of the comments.

However, issues with the data in Fig. 7 and Sup. Fig. 12 remain.

- In their rebuttal the authors wrote (regarding the model used in Fig. 7 and Sup. Fig. 12): “however, in some cases, tumor cells grew in the subarachnoid space outside the meninges causing a premature death of these animals. Thus, it is not possible to accurately determine the survival rate in the two conditions.” This comment raises questions about Sup. Fig. 12e, which directly compares the survival rate of these two conditions. Considering their comments, the authors should remove Sup. Fig. 12e from the manuscript. In addition, they should better explain the limitations of their model in the text (similar to what they did in the rebuttal).

We thank the Reviewer for the suggestion, and we have accordingly removed the Kaplan-Meier curve panel (Supplementary Figure 12e in the previous version) from the figure. We have also modified the Results section regarding this experiment including the text about this limitation (highlighted in green) previously only mentioned in the rebuttal letter.

- The q-RT PCR analysis in Figure 7f has only one biological replicate per group (n=1). If no additional biological replicates are available, please consider removing this panel. Alternatively, clearly indicate this limitation in the main text of the manuscript.

The Reviewer is correct: since we were able to collect CD11b⁺ cells only from two samples (one EV and one ZBTB18 FL), the qRT-PCR has only one biological replicate and the error bar in Figure 7f refers to the technical replicates (as stated in the figure legend). We have decided to keep this piece of data in the manuscript because both tested alleles are consistent, and it supports our previous staining and RNAseq data about MHCII. We have now clearly stated in the Results section referring to this experiment that only one biological replicate for each group was available and that the result is preliminary and requiring further confirmation.